# Liver and pancreatic-targeted interleukin-22 as a therapeutic for metabolic dysfunction-associated steatohepatitis

Haressh Sajiir [1,2,17], Sahar Keshvari [1,2,17], Kuan Yau Wong[1,2], Danielle J. Borg [1,2], Frederik J. Steyn [2], Christian Fercher [3,4], Karin Taylor[4], Breten Taylor[4], Ross T. Barnard [3,4], Alexandra Müller[1,2], Md Moniruzzaman [1,2], Gregory Miller[2,5], Ran Wang [1,2], Amelia Fotheringham[1,2], Veronika Schreiber[1,2], Yong Hua Sheng [1,2], Janelle Louise Hancock[6], Dorothy Loo[6], Lucy Burr[1,2,7], Tony Huynh [8,9,10,11], Jack Lockett [1,2,11], Grant A. Ramm [2,12], Graeme A. Macdonald [2,13], Johannes B. Prins[14], Michael A. McGuckin [15] & Sumaira Z. Hasnain [1,2,16] ✉

Metabolic dysfunction-associated steatohepatitis (MASH) is the most prevalent cause of liver disease worldwide, with a single approved therapeutic. Previous research has shown that interleukin-22 (IL-22) can suppress β-cell stress, reduce local islet inflammation, restore appropriate insulin production, reverse hyperglycemia, and ameliorate insulin resistance in preclinical models of diabetes. In clinical trials long-acting forms of IL-22 have led to increased proliferation in the skin and intestine, where the IL-22RA1 receptor is highly expressed. To maximise beneficial effects whilst reducing the risk of epithelial proliferation and cancer, we designed short-acting IL-22-bispecific biologic drugs that successfully targeted the liver and pancreas. Here we show 10-fold lower doses of these bispecific biologics exceed the beneficial effects of native IL-22 in multiple preclinical models of MASH, without off-target effects. Treatment restores glycemic control, markedly reduces hepatic steatosis, inflammation, and fibrogenesis. These short-acting IL-22-bispecific targeted biologics are a promising new therapeutic approach for MASH.

Metabolic dysfunction-associated steatotic liver disease (MASLD), previously known as Non-alcoholic Fatty Liver Disease (NAFLD)[1], encompasses a spectrum of disease from simple hepatic steatosis to metabolic dysfunction-associated steatohepatitis (MASH), is the most prevalent cause of liver disease worldwide[2]. Global Data forecasts that the total prevalence of MASH will grow from the current prevalence of ~62 M cases at a compound annual growth rate of 69% by 2029. Despite the extensive clinical trial pipeline, only one therapeutic[3] (Resmetirom; THR-β agonist) has been approved for MASH treatment to date. Most of the current agents in the pipeline only target 1 of the 4 key pathways activated in MASH: hepatic fat accumulation, hepatic inflammation,

upstream metabolic changes (glucose tolerance, insulin tolerance), and fibrosis[4].

We previously reported in preclinical murine models of diabetes, two-week administration of Interleukin-22 (IL-22) directly suppressed β-cell oxidative and endoplasmic reticulum (ER) stress, reduced local islet inflammation, restored appropriate insulin production, reversed hyperglycemia and ameliorated insulin resistance[5]. We noted additional beneficial changes with IL-22 treatment including body weight loss, increase in brown fat, reduction in ER stress in the liver, and improved intestinal barrier integrity[5,6]. The IL-22 receptor, IL-22RA1, is highly expressed in the pancreas, liver, intestine, and skin, with a role in

wound repair[7], but is absent on leukocytes. Administration of a long-acting IL-22-Fc fusion protein and recombinant IL-22 improved hyperglycemia in mouse models of obesity and reduced hepatic steatosis[8].

Despite demonstrated benefits, administration of IL-22 (particularly long-acting forms) may lead to detrimental unwanted effects. In preclinical models, IL-22 has been shown to increase cell proliferation in the skin and intestine[9,10]. Furthermore, in a phase I study of long-acting agents (dimerized IL-22 (F-652) and IL-22-Fc), 100% of the volunteers developed adverse events including eczematous lesions and/or gastrointestinal dysfunction[11]. To address the issue of off-disease-target effects, we designed short-acting IL-22-bispecific biologics that target the liver and pancreas; key organs in the development of MASH. Here we demonstrate that these IL-22-fusions target multiple pathways involved in MASH pathogenesis. We show that a 10-fold lower dose of these targeted biologics retain the beneficial effects of systemic native IL-22 in pre-clinical models of MASH. Furthermore, we demonstrate these novel IL-22 fusions target the pancreas and liver preferentially, without off target side-effects in the skin and intestine. Finally, we show these agents have a profound effect on preventing the development of hepatic fibrosis.

## Results

### Recombinant mouse IL-22 improved glucose tolerance, reduced weight, and induced satiety without altering metabolic rate or energy expenditure

Consistent with our previous results[5], biweekly recombinant mouse IL-22 (rmIL-22) treatment in the high-fat diet-induced obesity murine model induced a reduction in body weight which was accompanied by improved glucose tolerance (Supplementary Fig. 1a–c). We used the TSE Phenomaster system, to evaluate the effect of rmIL-22 on metabolic parameters in normal chow (NCD) and obese (high fat diet-fed, HFD) animals. Mean daily $VO_2$ was increased in HFD mice, but no effect of rmIL-22 treatment was observed, demonstrating that rmIL-22 did not affect metabolic rate; Supplementary Fig. 1d, f. Similarly, rmIL-22 had no impact on mean daily activity, Supplementary Fig. 1g, h.

No change in food intake was observed with rmIL-22 in NCD mice; Supplementary Fig. 1i–k. However, there was a significant reduction in cumulative food intake in obese mice treated with rmIL-22. Mechanistically, rmIL-22 significantly reduced the hypothalamic expression of genes known to enhance appetite (Neuropeptide-Y; NPY and Agouti-Related Peptide; AgRP) and promoted the expression of genes regulating satiety (pro-opiomelanocortin; POMC)[12]; Supplementary Fig. 1l. These data suggest that rmIL-22 treatment directly or indirectly improves regulatory pathways in the brain leading to improved satiety and an overall reduction in food intake.

### Long-term systemic IL-22 treatment reduced hepatic lipid accumulation but induced skin and intestinal inflammation and hyperproliferation

We have previously demonstrated rmIL-22 treatment reduced body fat content, which maybe be solely explained by the changes in satiety[5,8]. Therefore, to assess whether IL-22 could directly impact fat, we determined the changes in adipocyte size following rmIL-22 treatment, Fig. 1a. *IL-22ra1* expression was confirmed in epididymal and subcutaneous fat (Supplementary Fig. 2a), confirming the possibility of a direct effect. Obesity resulted in significant increase in adipocyte size of epididymal and subcutaneous fat cells compared to NCD, as previously reported[13]. rmIL-22 treatment significantly reduced adipocyte size in epididymal and subcutaneous fat tissue of HFD mice (Fig. 1b, c) but had no impact on adipocyte size in lean animals. Although there were significant alterations in the size of the adipocytes, no changes were observed in fasting serum total, High (HMW) or low molecular weight (LMW) adiponectin with HFD or with rmIL-22-treatment

(Supplementary Fig. 2b). Similarly, rmIL-22 significantly reduced the lipid accumulation in the liver of obese animals (Fig. 1d, e).

Whilst the beneficial effects in the lipid accumulation with rmIL-22 are desirable, previous studies using long-acting IL-22 (dimerized IL-22 (F-652) and IL-22-Fc) show adverse dose-related skin and intestinal effects[8,11,14]. We have confirmed these findings by treating C57BL/6 animals daily with increasing doses of recombinant mouse IL-22-Fc (rmIL-22-Fc, long-acting) for 8 weeks; Fig. 1f. An increase in the colon weight per length ratio, indicating inflammation and/or hyperproliferation, was observed in animals administered the highest dose of rmIL-22-Fc (Fig. 1g). Corroborating the clinical data, we observed over a doubling in epidermal thickness and proliferation in the rmIL-22-Fc treated animals in the injected and un-injected sites (Fig. 1h, i). Moreover, an increase in epidermal neutrophil infiltration was seen in the animals administered rmIL-22-Fc compared with PBS controls (Fig. 1i). No histological changes were observed in the liver or in the exocrine or endocrine pancreas after rmIL-22-Fc treatment (Supplementary Fig. 3).

### Designing liver/pancreas targeted IL-22-bispecific biologics

The long-circulating IL-22 fusions improved MELD (Model of End-Stage Liver Disease) and Lille scores in patients with alcoholic hepatitis following twice weekly *i.v.* administration of F-652 for 6 weeks, with pruritus as the most common adverse event[14]. We hypothesized that liver-targeted IL-22 would provide greater efficacy and/or improved therapeutic index. GLP1 (glucagon-like peptide) receptor binding proteins and single chain variable fragment (ScFv), known to target pancreas and/or liver, were used as targeting moieties in our novel IL-22-bispecific biologics, Supplementary Fig. 4a. Western blot and HPLC SEC analyses showed that human IL-22 (hIL-22) and hIL-22-GLP1 formed dimers and the hIL-22-ScFv fusions constructed in both N- and C-terminal fusions tended to form higher order oligomers (Supplementary Fig. 4b–d). Backscatter analyses suggested that the ScFv-hIL-22 fusion aggregated at higher temperatures and therefore maybe less stable in vivo (Supplementary Fig. 4e, f).

MIN6N8 insulinoma-derived beta cells and hepatocellular carcinoma cell lines, HEPG2 were used to assess the activity of the bispecific drugs and confirm retention of IL-22RA1 activation. IL-22RA1 transmits signals via the JAK/STAT pathway activating STAT3 via phosphorylation[8]. All fusions resulted in STAT3p activation 30 min post-stimulation, however activation was lower compared to native hIL-22; Supplementary Fig. 5a, b. There was a difference in the kinetics of STAT3-activation by the fusion proteins compared with native hIL-22. ScFv-hIL-22 only activated STAT3p in the HEPG2 cell lines, no STAT3p was observed in the MIN6N8 cells; Supplementary Fig. 5d, h. hIL-22-GLP1 and hIL-22-ScFv induced lower but sustained pSTAT3 activation in the HEPG2 cells compared with native hIL-22 (Supplementary Fig. 5i, j).

The sustained activity of hIL-22-GLP1 and hIL-22-ScFv could potentially be of benefit in-vivo, and therefore we adopted pStat3 as a practical in vivo bioassay to assess the tissue specificity of IL-22-based biologics (Fig. 2a). We observed significantly lower pStat3 activation in the intestine (-10-fold) with both hIL-22-ScFv and hIL-22-GLP1 (Fig. 2b–e), compared to native hIL-22. However, whilst a significant increase in targeting was observed to both the liver (-80-fold) and pancreas (-10-fold) with hIL-22-ScFv, hIL-22-GLP1 only enhanced targeting to the liver with pancreatic targeting remaining comparable to native hIL-22 (Fig. 2 b, c, e). hIL-22-ScFv significantly reduced tunicamycin-induced ER stress measured by *sXBP1* in human islets, confirming the retention of function similar to native hIL-22, and its direct effect on pancreatic islets. This reduction in ER stress was also accompanied by an improvement in tunicamycin-induced increased proinsulin levels in the human islets, a marker of beta-cell dysfunction[15,16]. ScFv-hIL-22 and hIL-22-GLP1 had lesser effects on tunicamycin-induced ER stress and alterations in proinsulin that did not reach statistical significance (Fig. 2f).

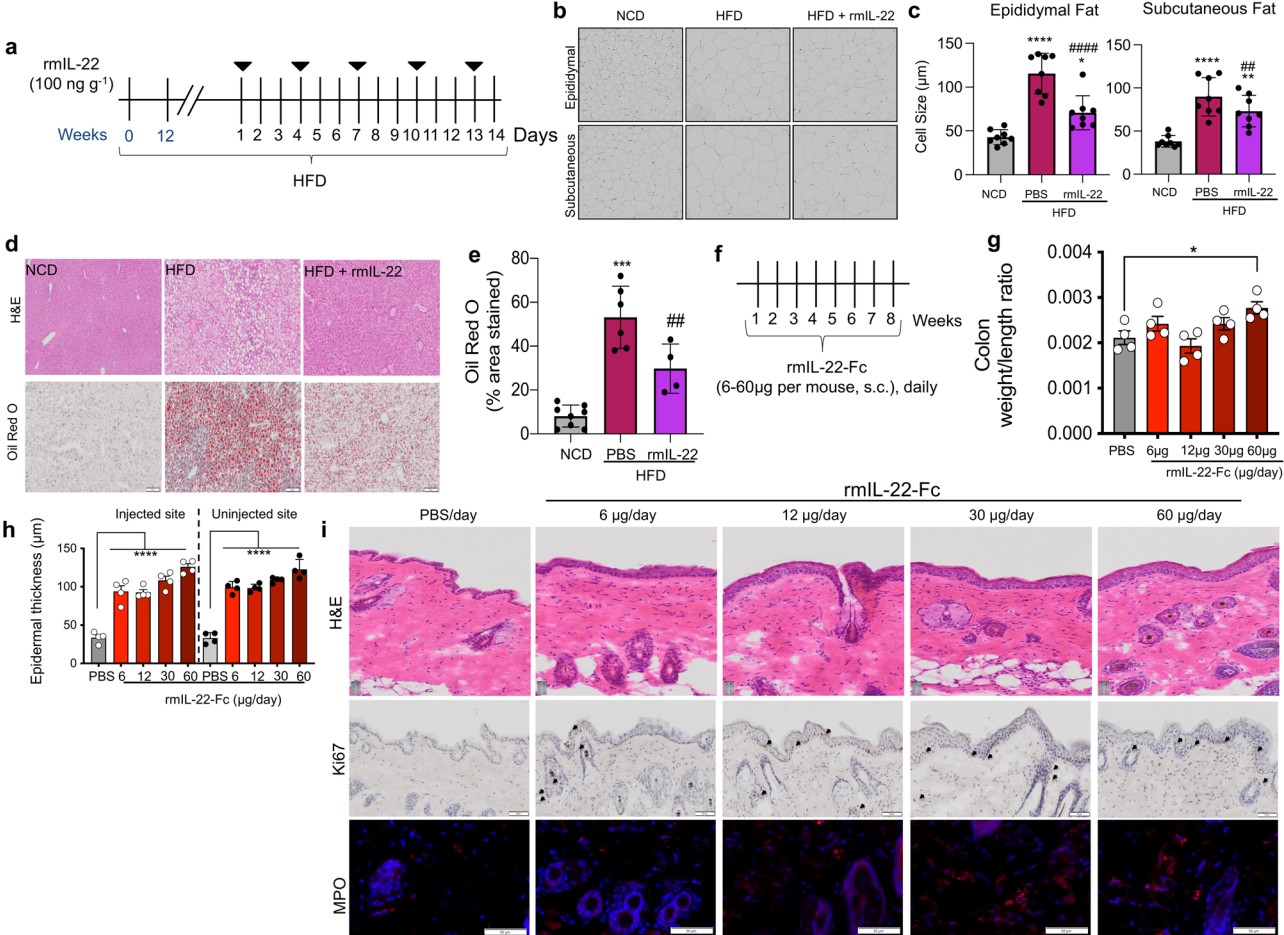

**Fig. 1 | Recombinant mouse IL-22 (rmIL-22) treatment reduces lipid accumulation, but extended treatment with long-acting forms can lead to hyperproliferation in the skin and intestine. a–e** mice were fed a normal chow diet (NCD) or a high fat diet (HFD) for 14 weeks and treated biweekly for the last two weeks with 100 ng g⁻¹ of recombinant mouse IL-22 (rmIL-22), i.p. Control NCD and HFD animals received PBS. **a** Experimental schematic. **b** Representative images of epididymal and subcutaneous fats stained with hematoxylin and eosin (H&E). **c** Average size of 100 adipocytes was measured in each section of the fat tissues. **d** Oil Red O staining and H&E staining of liver. **e** graph shows the % area stained with Oil Red O in the liver. **f–i** C57BL/6 animals were treated with either PBS or 6, 12, 30, or 60 µg of rmIL-

22-Fc daily for 8 weeks (s.c.). **f** Experimental schematic. **g** Ratio of colon weight:length as a measure of inflammation. **h** Quantification of epidermal thickness in µm from the un-injected and injected sites. **i** H&E. Ki67 and MPO staining of skin. **a–e** $n = 8$ biologically independent animals, ANOVA, Bonferroni's post hoc test. **$p < 0.01$ ****$p < 0.0001$ compared to NCD mice and ##p < 0.01 ####$p < 0.0001$ compared to control HFD. **f–i** $n = 4$ biologically independent animals. One-way ANOVA, Bonferroni's post hoc test. Data are presented as mean values ± SEM. *$p < 0.05$, ****$p < 0.0001$ compared to PBS-treated controls. Scale bar = 50 µm. Source data are provided as a Source Data file.

Biweekly treatment of obese animals (12 weeks of HFD) with hIL-22-ScFv or hIL-22-GLP1 for 2 weeks reduced the serum proinsulin to insulin ratio, indicating reduced beta-cell workload[5] (Fig. 2g, h). However, at the two-week treatment time point, in contrast to our previous observations with mouse rIL-22[5], human rIL-22 had no significant effect in improving proinsulin-insulin ratio (Fig. 2g). We postulate this is related to 79% species homology with reduced binding efficacy of human rIL-22 to mouse IL-22ra1. Human rIL-22 was approximately 25% less effective than mouse rIL-22 at alleviating palmitate-induced ER stress in a mouse cell line, which mimics the beta-cell dysfunction observed in obesity caused by elevated circulating free fatty acids that trigger cellular stress[5,17] (Supplementary Fig. 6a). The greater improvement in proinsulin:insulin ratio likely relates to enhanced pancreatic pStat3 activation with targeted bispecific fusions compared to native hIL-22; Fig. 2b, e. hIL-22-ScFv was the only compound evaluated that significantly improved fasting blood glucose (Fig. 2h), consistent with enhanced targeting to the liver in addition to the pancreas (Fig. 2b–e).

## Dual-targeting of IL-22-ScFv to pancreas and liver enhances the efficacy of IL-22 in improving metabolic syndrome

Based on these in vitro and in vivo comparisons of targeting and efficacy, we selected hIL-22-ScFv as the lead candidate for further development and comprehensive pre-clinical testing. Two additional variants of hIL-22-ScFv were prepared with differing linkage sequences (Supplementary Fig. 7a) and all hIL-22-ScFv variants retained pSTAT3 activation in HEPG2 and MIN6N8 cells (Supplementary Fig. 7e). Alteration of the linker length from (G4S)₁ to (G4S)₃ or (G4S)₅ had no significant impact on the pharmacokinetic profile, with near complete clearance of all variants by 8 h (Supplementary Fig. 7b). Targeting profile was not improved with any tested variation in linker length (Supplementary Fig. 7c, d). All variants (human and mouse) targeted the liver/pancreas, however variants with (G4S)₃ and (G4S)₅ had targeting to the intestine as well. Storage temperature sensitivity studies using in vitro STAT3p activation to assess activity showed hIL-22-ScFv was significantly more stable than native hIL-22, (Supplementary Fig. 8a). Surface plasmon resonance binding analyses confirmed hIL-22-ScFv bound (1:1) to the IL-22RA1 receptor (Supplementary

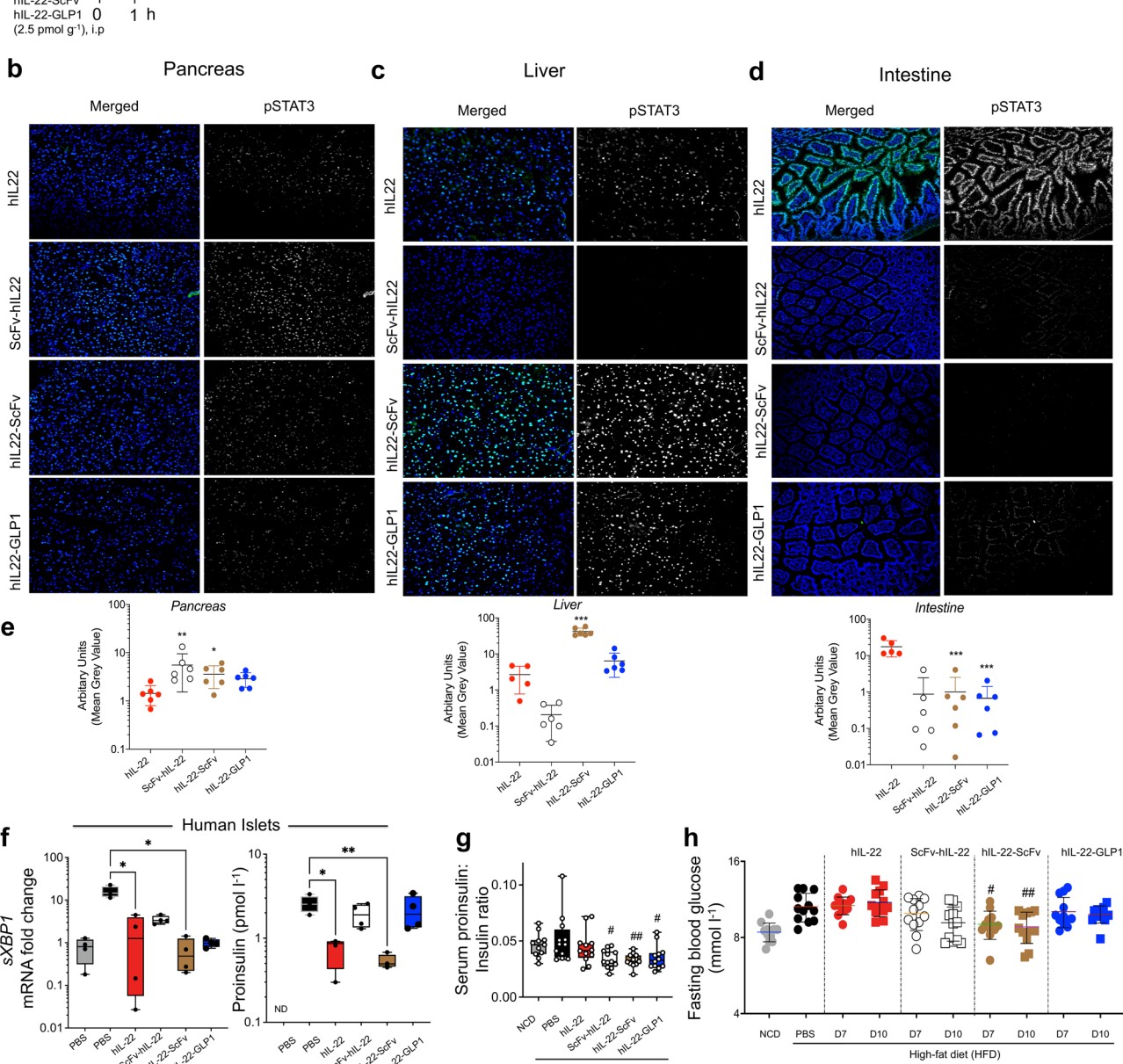

**Fig. 2 | Human IL-22-fusion drugs can enhance targeting to pancreas and liver.** **a–d** C57BL/6 animals were treated with native human IL-22 (hIL-22) or 3 different hIL-22-fusions at 2.5 pmol g⁻¹, i.p and sampled 1 h post-treatment. **a** Experimental schematic. Phospho-Stat3 staining analyzed by confocal microscopy in (**b**) pancreas, (**c**) liver and (**d**) intestine, (**e**) quantification of fluorescence staining intensity is shown as arbitrary units. (**f**) mRNA expression of ER stress marker (*spliced XBP1; sXBP1*) and protein levels of proinsulin in human islets cultured for 24 h (controls) or treated with a N-glycosylation inhibitor (tunicamycin) in the presence of native hIL-22, ScFv-hIL-22, hIL-22-ScFv and hIL-22-GLP1 (3 pmol mL⁻¹). ND = not detected. **g** Mice were fed a normal chow diet (NCD) or a high fat diet (HFD) for 14 weeks and treated biweekly for the last two weeks with native hIL-22 or hIL-22-fusions (ScFv-hIL-22, hIL-22-ScFv, hIL-22-GLP1), 2.5 pmol g⁻¹, i.p. Control NCD and HFD animals received PBS. Fed serum proinsulin:insulin ratio on day 14 post-treatment. **h** Fasting blood glucose on day 7 (D7) and day 10 (D10) post-treatment. **a–e** $n = 5–6$, One-way ANOVA, Bonferroni's post hoc test. *$p < 0.05$, **$p < 0.01$, ***$p < 0.001$, ****$p < 0.0001$ compared to mice given hIL-22. **f** $n = 4$ biologically independent human islet donors (**g–h**) $n = 12$ biologically independent animals. One-way ANOVA, Bonferroni's post hoc test. #$p < 0.05$, ##$p < 0.01$ ####$p < 0.0001$ compared to control HFD. Source data are provided as a Source Data file.

Fig. 8b, c). The kinetic analyses confirmed that binding and affinity was similar for all variants (human and mouse); Supplementary Fig. 8c.

Our lead bispecific compound, hIL-22-ScFv retained efficacy following i.p. administration, demonstrated by significant improvement in glycemic control and hepatic steatosis, inflammation and ER stress

(Fig. 3a-g). While fasting blood glucose levels remained largely unchanged after treatment with native IL-22 (Fig. 2h), oral glucose tolerance improved. This suggests that overall glucose handling becomes better before fasting blood glucose levels change. These changes were accompanied by an increase in brown fat pads and an

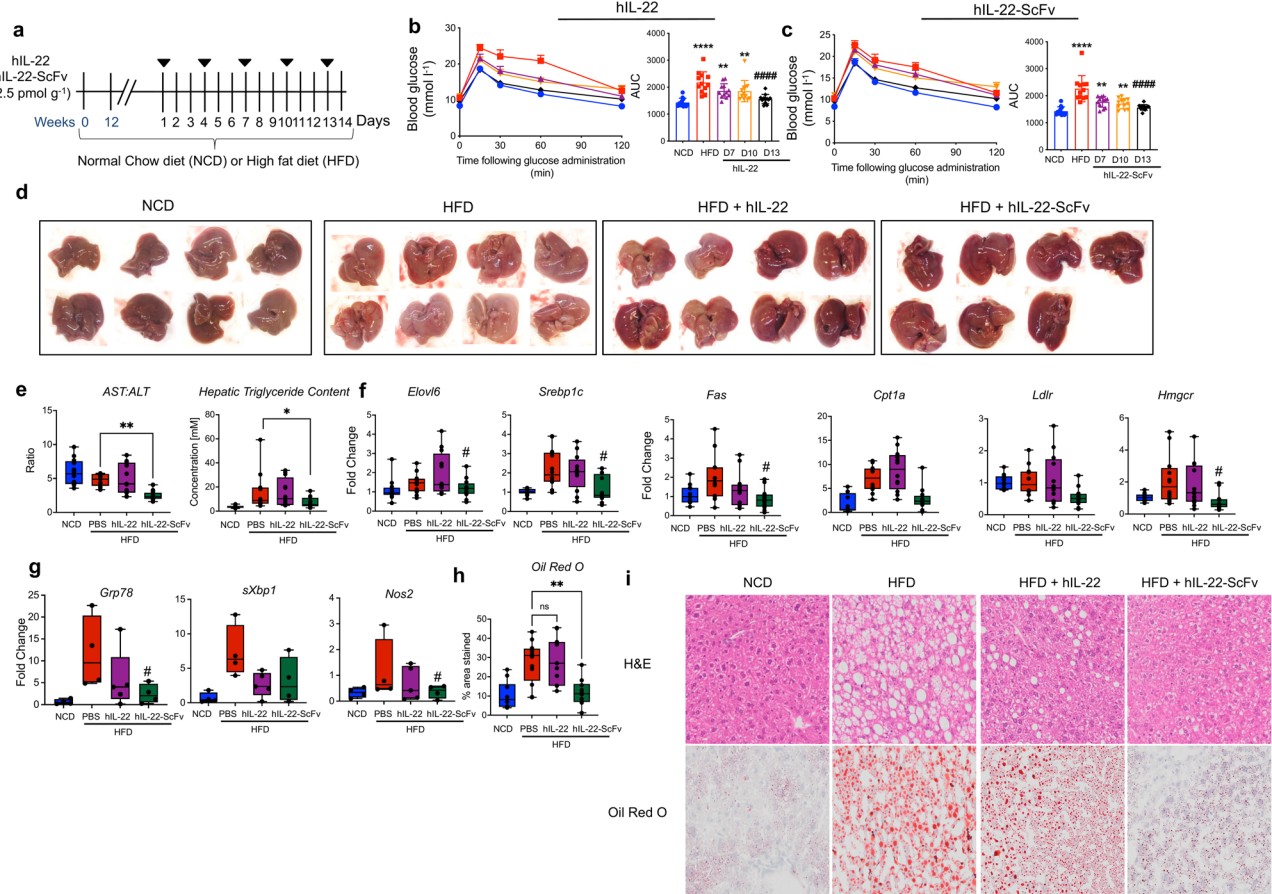

**Fig. 3 | Targeted IL-22 improves glucose tolerance and hepatic lipid accumulation. a** Mice were fed a normal chow diet (NCD) or a high fat diet (HFD) for 14 weeks and treated biweekly for the last two weeks with 2.5 pmol g⁻¹ of native hIL-22 or hIL-22-ScFv, i.p. Control NCD and HFD animals received PBS. Oral glucose tolerance test on day 7, 10, 13 post-treatment of animals treated with (**b**) hIL-22 or (**c**) hIL-22-ScFv. AUC for each treatment shown. **d** Gross macroscopic images of the livers from NCD, HFD and HFD animals treated for 2 weeks with hIL-22 or hIL-22-ScFv. **e** Serum AST:ALT ratio, hepatic triglyceride content (presented as concentration) in homogenized liver (normalized to weight of liver). **f** Hepatic mRNA expression of genes associated with lipid biosynthesis, and (**g**) ER stress markers *Grp78* and *spliced-Xbp1* (*sXbp1*) and oxidative stress marker *Nos2*. **h** Quantification of Oil Red O staining shown as percentage area stained. **i** Hematoxylin and Eosin (H&E) and Oil Red O staining of livers 2 weeks post-treatment. $n = 8$ biologically independent animals for NCD, HFD/hIL-22 and HFD/hIL-22-ScFv groups, and $n = 10$ biologically independent animals for the HFD/PBS group. One-way ANOVA, Bonferroni's post hoc test. Bar graphs in (**b**, **c**) are presented as mean values ± SEM. #/*$p < 0.05$, ##/**$p < 0.01$ ####/***$p < 0.0001$ compared to control HFD or NCD, respectively. Source data are provided as a Source Data file.

overall a slight decrease in fat mass in the animals treated with hIL-22-ScFv but not native hIL-22 (Supplementary Fig. 6b, c), suggesting that targeted-IL-22 (hIL-22-ScFv) has increased efficacy compared to native hIL-22 when administered intraperitoneally. HFD animals had gross macroscopic features of hepatic steatosis, which were improved with hIL-22-ScFv treatment (Fig. 3d). Consistent with the changes in liver gross morphology, hIL-22-ScFv treatment reduced the AST/ALT ratio, and significantly reduced triglyceride content and lipid accumulation in the liver (Fig. 3e, h, i). The hepatic expression of ER stress marker *Grp78* and oxidative stress marker *Nos2* was also significantly reduced with hIL-22-ScFv treatment (Fig. 3g). Paralleling the decrease in cellular stress, key enzyme genes important in lipid metabolism (*Elovl6, Srebp1c, Fas, Hmgcr*) were downregulated in the HFD animals treated with hIL-22-ScFv but remained unaltered with hIL-22 treatment (Fig. 3f).

## Targeting to liver and pancreas is maintained with subcutaneous administration of IL-22-ScFv

With route of administration in humans in mind, we sought to ensure that IL-22-ScFv can be administered subcutaneously in animals whilst retaining its efficacy. Subcutaneous administration of recombinant mIL-22-ScFv at 12.5 pmol g⁻¹ reached the same peak concentration

(~8 ng mL⁻¹) as the intraperitoneal dose shown to be efficacious (2.5 pmol g⁻¹); Supplementary Fig. 8d. We confirmed that targeting was maintained even with high doses of mIL-22-ScFv (400 pmol g⁻¹) when administered subcutaneously. pStat3 activation was apparent in the liver and pancreas, with negligible staining observed in the kidney, heart, lung, spleen, and gut. As expected with the subcutaneous injection, with high doses of mIL-22-ScFv some Stat3p in epidermal cells was observed at the injection site, however no epidermal Stat3p was observed in un-injected skin; Supplementary Fig. 8e.

Subcutaneous delivery of mIL-22-ScFv (12.5 pmol g⁻¹ or 6.25 pmol g⁻¹) was effective, in improving hyperglycemia and hepatic lipid accumulation, when given weekly or biweekly to obese animals (Supplementary Fig. 9a). There were no significant changes in the body weight, fat mass or brown fat pads (Supplementary Fig. 9b), when mIL-22-ScFv was administered subcutaneously. 6.25 pmol g⁻¹ weekly or biweekly consistently decreased hyperglycemia, whereas a subset of animals in the 12.5 pmol g⁻¹ treated group remained glucose intolerant; area under the curve shown in Supplementary Fig. 9c. 6.25 pmol g⁻¹ was also more effective than the higher dose of 12.5 pmol g⁻¹ in reducing hepatic lipid, accompanied by a reduction in hepatic inflammatory cytokines (*Il1β and Tnfα*) and fatty acid synthase (*Fas*) expression (Supplementary Fig. 9d, e).

To assess the long-term effects of mIL-22-ScFv on skin and gut we treated obese animals (12 weeks HFD) for 8 weeks with mIL-22-ScFv or mouse IL-22-Fc (mIL-22-Fc) as a comparator (Supplementary Fig. 10a). A marked increase in the small intestinal (ileum) villus length, accompanied by an increase in Ki67[+ve] staining was observed with mIL-22-Fc treatment, which was not observed with mIL-22-ScFv treatment (Supplementary Fig. 10b–d). Additionally, mIL-22-ScFv high dose did not induce the thickening of the epidermis either at the injected or uninjected site, which was evident at both sites in the animals treated with mIL-22-Fc (Supplementary Fig. 10e, f). Notably, there was a significant increase in Ki67[+ve] cells in the injected site of mIL-22-Fc treated animals, accompanied by an increase in TUNEL staining suggesting an increase in both cell division and death at the injection site with mIL-22-Fc; Supplementary Fig. 10e.

RT²PCR Profiler array revealed that hIL-22-ScFv treatment in obese animals enhanced the hepatic expression of *Socs3* (suppressor of cytokine signaling-3) that is a cytokine driven STAT-induced suppressor of cytokine signaling and *Pck2* (Phosphoenolpyruvate carboxykinase-2) an enzyme that in the liver regulates gluconeogenesis. Concomitantly, hIL-22-ScFv downregulated genes associated with inflammation, including *Casp3* (Caspase-3), *Ifnγ* (Interferon-gamma) and *Il6* (Fig. 4a; Supplementary Fig. 11a). Interestingly, the most highly altered genes with hIL-22-ScFv were downregulation of beta-2-microglobulin (*B2m*) and pyruvate dehydrogenase kinase-4 (*Pdk4*), which have both been implicated in driving fibrosis (Fig. 4a, Supplementary Fig. 11a).

### Targeted IL-22-ScFv is more effective in reducing hepatic fibrosis compared with long-acting IL-22

We next investigated whether IL-22-ScFv would prevent the transition to fibrosis in multiple models of MASH. The choline-deficient, L-amino acid-defined, high-fat diet (CDAHFD) model was first assessed. CDAHFD mice were treated with mIL-22-ScFv or mIL-22-Fc for 5 weeks while being kept on the CDAHFD (Fig. 4b), and no changes were noted in body weight, fasting blood glucose, or the weight of pancreas or liver (Supplementary Fig. 11b). There was a non-significant trend toward a reduction in lipid accumulation; however, a significant decrease in lipid size was observed in mIL-22-ScFv but not mIL-22-Fc treated animals (Fig. 4c, d). This was paralleled by a significant decrease in NAS and fibrosis scores with mIL-22-ScFv, whereas mIL-22-Fc reduced NAS and fibrosis scores in a minor subset of mice only (not statistically significant); Fig. 4d. The decrease in fibrosis with mIL-22-ScFv was also apparent with Sirius red and Masson's trichrome staining and supported by the reduction in the expression of pro-fibrotic collagen genes including *Col1a2* (collagen type-1 alpha-2 chain) and *Col3a1* (collagen type-3 alpha-1 chain) (Fig. 4e, f). The enhanced efficacy of mIL-22-ScFv, at a tenth of the molar dose of mIL-22-Fc, could be due to the enhanced targeting of the liver with mIL-22-ScFv and/or their different pharmacokinetic profiles. RT²PCR Profiler array analysis revealed that a large number of genes in pro-fibrogenic and inflammatory pathways activated under CDAHFD were significantly and substantially reduced in expression by mIL-22-ScFv treatment (Supplementary Fig. 11c, d).

Next, obese animals were co-administered HFD and thioacetamide (HFD/TAA) for 16 weeks and treated in the last 5 weeks with mIL-22-Fc or mIL-22-ScFv (Supplementary Fig. 11e, f). H&E, Sirius red and Masson's Trichrome staining confirmed severe diffuse fibrosis, hepatic inflammation, and collagen deposition in this model (HFD/TAA, PBS) as previously reported[18] (Fig. 4g). A significant reduction in fibrosis was observed with mIL-22-ScFv but not mIL-22-Fc treatment (Fig. 4h) and microarray analyses demonstrated a reduction in pro-fibrotic/inflammatory genes *Mmp3*, *Tgfb2* and *Timp4* with mIL-22-ScFv (Supplementary Fig. 11g). In this model mIL-22-Fc but not mIL-22-ScFv increased liver weight ~30% above untreated and HFD/TAA control mice, indicative of inflammation and/or hyper-proliferation (Supplementary Fig. 11f).

### Pancreatic and hepatic IL-22RA1 signaling are both required for metabolic improvements with IL-22-ScFv treatment

Given the multiple effects of targeted IL-22-ScFv in improving hyperglycemia, hepatic inflammation, steatosis, and fibrosis, we generated pancreatic beta-cell specific *IL-22ra1* knockout mice (Ins2-Cre x *IL-22ra1*[fl/fl]) hereafter referred to as *IL-22ra1*[βcell−/−], to dissect the tissue-specific effects of mIL-22-ScFv. *IL-22ra1*[βcell−/−] mice kept on a HFD for 12 weeks were treated with mIL-22-ScFv. mIL-22-ScFv treated wild-type animals (*IL-22ra1*[fl/fl]) showed an improvement in hyperglycemia. No improvements in hyperglycemia or proinsulin/insulin levels were observed in these animals when treated with mIL-22-ScFv, confirming that *IL-22ra1* β-cell signaling is critical for the improvements in glucose control (Fig. 5a). Whilst mIL-22-ScFv reduced hepatic lipid accumulation in the *IL-22ra1*[βcell−/−] fed a HFD, this effect was lost in the *IL-22ra1*[Hep−/−] HFD-fed mice (Alb-Cre x *IL-22ra1*[fl/fl]). These data demonstrate that reduced steatosis is independent of altered β-cell function and glycemic control and likely that IL-22RA1-hepatic signaling is required for IL-22-ScFv-driven improvements in the liver (Fig. 5b–e).

### Multimodal effects of IL-22-ScFv in MASH

We hypothesized that IL-22 can directly affect lipid storage in hepatocytes, therefore we treated HEPG2 cells with palmitate, inducing an increase in lipid accumulation in a dose dependent manner (Supplementary Fig. 12a). Treatment of lipid-containing HEPG2 with hIL-22-ScFv demonstrated a clear overall decrease in lipid storage using Bodipy staining (Fig. 5f, g). We conducted gene expression analyses, which highlighted some subtle but significant increases in *carnitine palmitoyltransferase-1a* (*CPT1a*), and *acetyl-coA carboxylase-alpha* (*ACACA*) *gene* (Supplementary Fig. 12b). While this suggested that IL-22-ScFv was altering lipid storage, the pathways altered remained unclear. To investigate this further, and analyze the global changes associated with decrease in lipid storage, we conducted mass spectrometry proteomics on lipid-containing HEPG2 cells with and without hIL-22-ScFv treatment. This unbiased approach resulted in identifying 5480 proteins with high confidence, with 250 differential proteins identified (with $p < 0.05$; multiple peptides identified); IL-22 was identified in the IL-22-ScFv treated sample serving as an internal control (Supplementary Fig. 12c). DAVID Bioinformatics analysis, specifically focusing on KEGG pathway enrichment, revealed that 36% of the differentially expressed proteins are directly associated with metabolic pathways. Of significance is also function within the endoplasmic reticulum (ER), among the upregulated pathways known to be regulated by IL-22-ScFv. However, it's important to note that these altered pathways encompass various aspects of lipid and fatty acid metabolism, suggesting a broad impact on the cell's metabolic state. Therefore, these data demonstrate a complex modulation of metabolic pathways and may be one of the mechanisms by which IL-22-ScFv reduces lipid storage in hepatocytes (Fig. 5h).

hIL-22-ScFv significantly reduced fibrosis in vivo, and hIL-22-ScFv and its variants all activated pSTAT3 in LX2 human stellate cells. This indicates that hIL-22-ScFv can directly target stellate cells, the primary cell type responsible for fibrosis development (Supplementary Fig. 12d). To confirm a direct IL-22-ScFv-driven effect on the stellate cells in-vivo, CDAHFD animals were pulsed with 400 pmol g⁻¹ of mIL-22-ScFv with immunofluorescence staining of pStat3 and stellate cell markers (GFAP and Reelin) confirmed the targeting of stellate cells (Fig. 5i). Activated stellate cells are responsible for driving fibrosis and the accumulation of type-I collagen in the diseased liver and direct treatment with hIL-22-ScFv reduced *ACTA2, COL1A3, TGFβ2* in the LX2 cells. Moreover, an increase in *SOCS3* (Fig. 5j), suggested that IL-22-ScFv, as shown previously for native IL-22, can induce senescence and

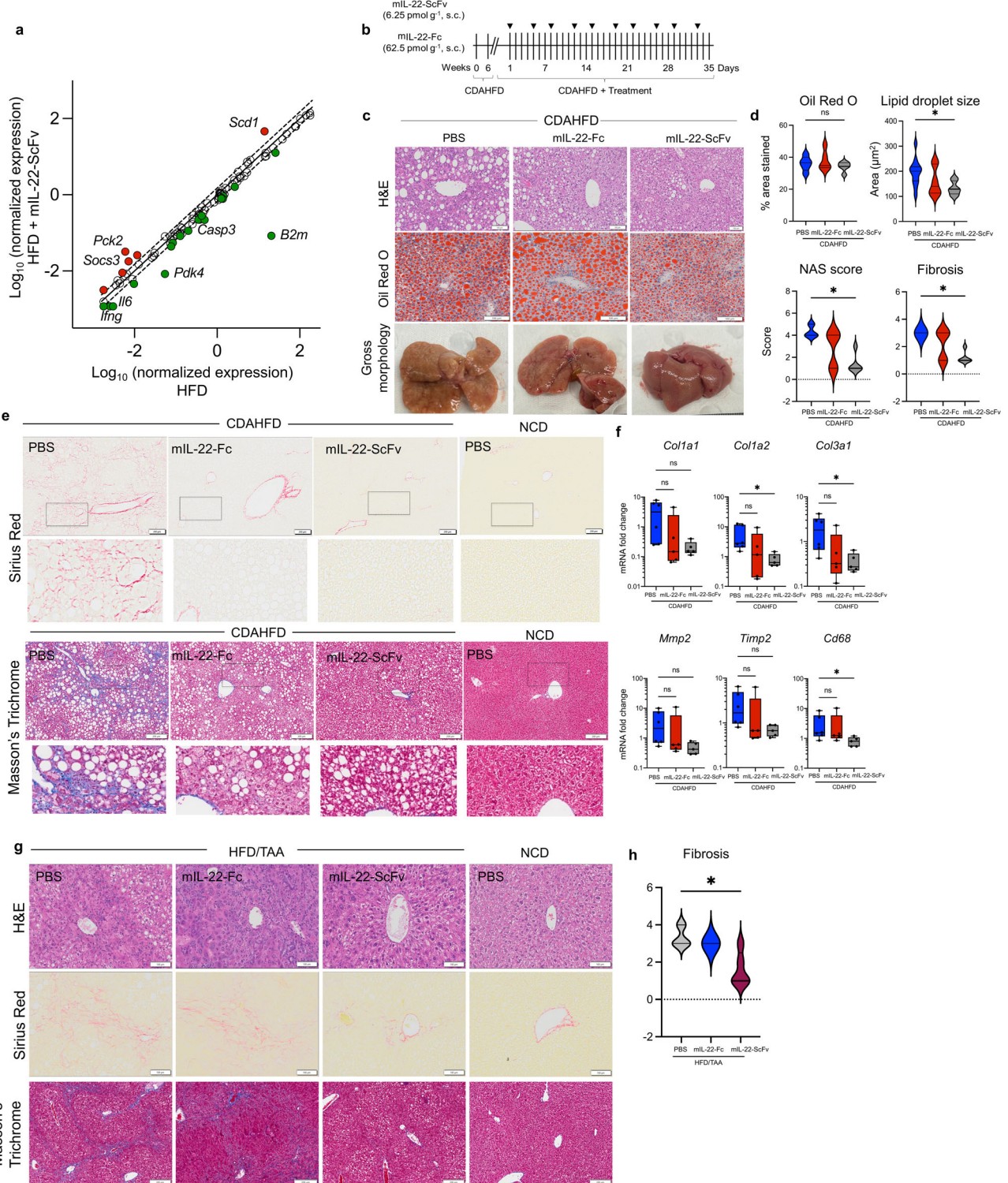

**Fig. 4 | Murine IL-22-ScFv stops the progression of hepatic fibrosis. a** Fatty Liver RT$^2$ Profiler PCR Array on HFD (12 weeks) and HFD animals treated with 6.25 pmol g$^{-1}$ mIL-22-ScFv (2 weeks), $n = 2$ biologically independent animals. **b** Schematic showing C57BL/6 animals placed on a choline-deficient, L-amino acid-defined, high-fat diet (CDAHFD) for 6 weeks and treated biweekly with mIL-22-Fc (62.5 pmol g$^{-1}$) or mIL-22-ScFv (6.25 pmol g$^{-1}$) for 5 weeks. **c** End-point histology assessment, H&E, Oil Red O staining and gross morphology representative images. **d** shows the % area stained with Oil Red O, image J analyses of lipid droplet size, NAS scoring and fibrosis scores. **e** Sirius Red and Masson's Trichrome staining on CDAHFD animals with and without treatment. **f** RT-PCR was used to confirm the improvements in fibrosis markers and inflammation. $n = 6$ biologically independent animals for the CDAHFD/PBS group,

$n = 5$ biologically independent animals for the NCD, CDAHFD/mIL-22-ScFv and CDAHFD/mIL-22-Fc groups. **g** Obese C57BL/6 animals were placed on an HFD/TAA (regime shown in Supplementary Fig. 11e) and treated with mIL-22-Fc or mIL-22-ScFv. H&E, Sirius Red and Masson's Trichrome staining of representative images shown. **h** shows the fibrosis scores. $n = 3$ biologically independent animals for the NCD group, $n = 4$ biologically independent animals for the HFD/TAA/PBS group, $n = 5$ biologically independent animals for the HFD/TAA/mIL-22-Fc and HFD/TAA/mIL-22-ScFv groups. One-way ANOVA, Bonferroni's post hoc test. *$p < 0.05$, **$p < 0.01$, ***$p < 0.001$, ****$p < 0.0001$ compared to HFD control. Source data are provided as a Source Data file.

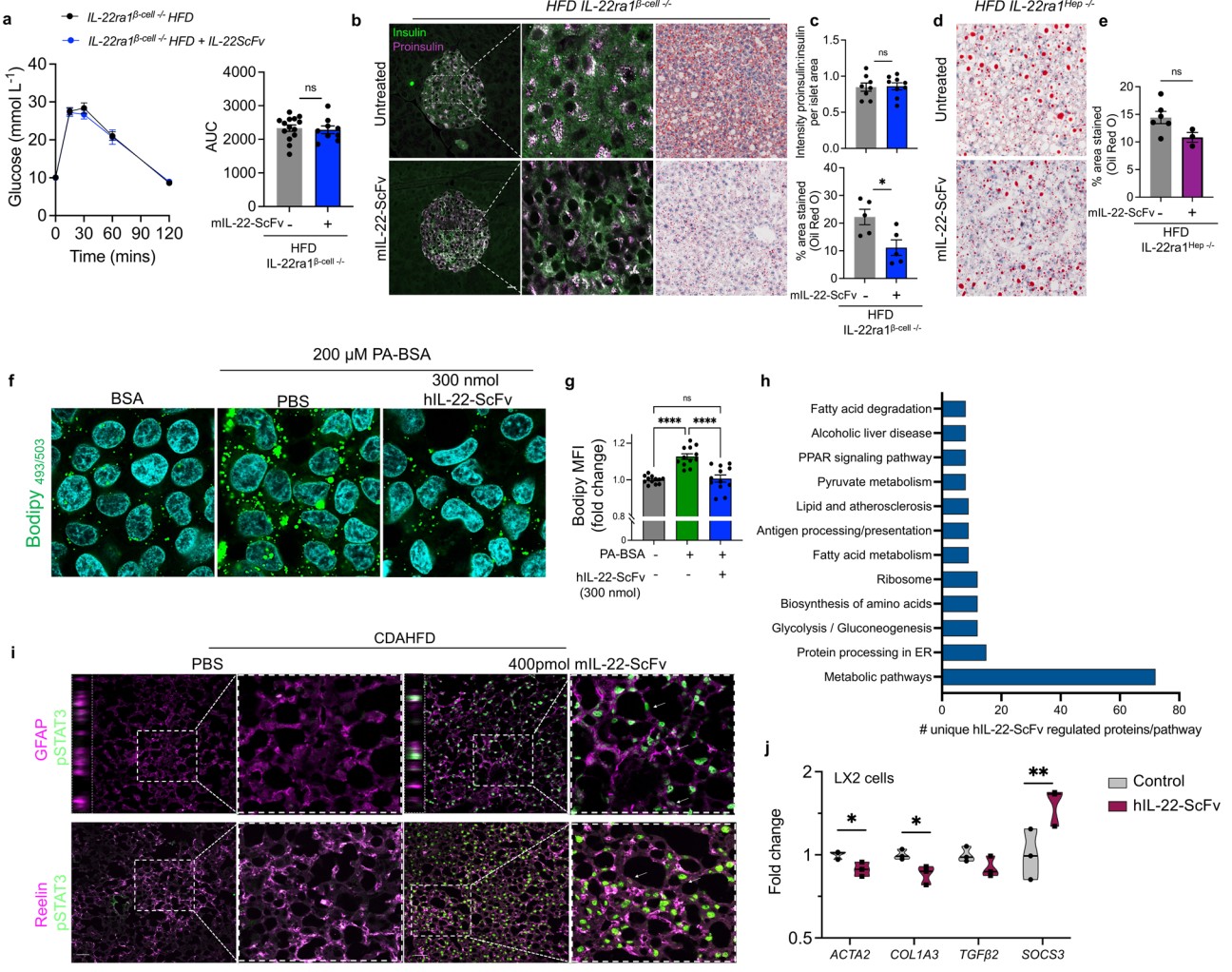

**Fig. 5 | IL-22-ScFv acts through hepatic IL-22RA1 signaling in hepatocytes and stellate cells to provide multiple benefits in MASLD/MASH. a** Oral glucose tolerance test in animals with an IL-22ra1 β-cell specific deletion *IL-22ra1^β-cell-/-^*; animals were put on a HFD for 12 weeks and treated biweekly with 6.25 pmol g⁻¹ mIL-22-ScFv or PBS (s.c.) for the last 2 weeks of diet. *n* = 14 biologically independent animals in the control group, and *n* = 9 biologically independent animals in the *IL-22ra1^β-cell-/-^* group. **b** Insulin and Proinsulin staining of pancreas, and Oil red O staining of liver in the animals in (**a**). **c** Quantitation of proinsulin:insulin ratio and Oil red O area stained. *n* = 8 biologically independent animals in control islet staining group, *n* = 9 biologically independent animals in *IL-22ra1^β-cell-/-^* islet staining group; *n* = 5 biologically independent animals in Oil red O staining groups. **d** Animals with a hepatocyte specific deletion of IL-22ra1, *IL-22ra1^Hep-/-^*, were put on a HFD for 12 weeks and treated biweekly with 6.25 pmol g⁻¹ mIL-22-ScFv or PBS (s.c.) for the last 2 weeks of diet, liver tissue stained with Oil Red O. **e** Quantitation of Oil Red O area stained. *n* = 6 biologically independent animals in the control group, *n* = 3 biologically independent animals in the *IL-22ra1^Hep-/-^* group. **f, g** HEPG2 cells treated with

Palmitate (200 μM) to induce lipid accumulation and treated with hIL-22-ScFv (300 nmol mL⁻¹); (**f**) representative images and (**g**) mean fluorescence intensity (MFI) shows lipid accumulation using Bodipy™ ₄₉₃/₅₀₃ (shown as fold change of BSA controls). *n* = 12 independent samples and (**h**) Mass Spectrometry and KEGG pathway analyses of differentially expressed unique proteins in lipid-containing HEPG2 cells treated with hIL-22-ScFv compared with control. *n* = 3 for BSA controls; *n* = 4 for PA-BSA ± hIL-22-ScFv. **i** CDAHFD animals were pulsed with a single dose of 400 pmol g⁻¹ of mIL-22-ScFv (s.c), liver tissue stained with STAT3p and, GFAP and Reelin to identify stellate cells (white arrows); inset = z-stack. *n* = 4 biologically independent animals. Scale bar = 50 μm. **j** LX2 cells were activated using hTGF-beta and subsequently treated with 300 nmol mL⁻¹ hIL-22-ScFv, gene expression fold change in *ACTA2, COL1A3, TGFbeta2, SOCS3* compared to control. *n* = 3 independent samples. T-test or One-way ANOVA, Bonferroni's post hoc test. Bar graphs in (**a, c, e, g**) are presented as mean values ± SEM. *\*p* < 0.05, *\*\*p* < 0.01, *\*\*\*p* < 0.001, *\*\*\*\*p* < 0.0001. scale bar = 100 μm. Source data are provided as a Source Data file.

anti-inflammatory pathways in stellate cells[19]. Therefore, our data suggests that the anti-fibrotic effects of IL-22-ScFv were a combinatory effect of treatment on the different cell types in the liver, including lipid-containing hepatocytes, and stellate cells and pancreatic beta cells.

## Discussion

Approximately 38% of the global population is estimated to have MASLD, which is forecasted to increase by 50% in the next 10 years[20,21]. Despite the high incidence and potential severity, there is currently a paucity of therapeutic agents for the treatment of MASLD/MASH. There are several drugs in the MASH clinical pipeline which typically

only target one of the activated pathways contributing to MASH pathology. Here we demonstrate using multiple preclinical models of MASH that pancreas- and liver-targeted IL-22 beneficially modulates multiple pathways, including reducing hyperglycemia by improving beta cell function and quality of secreted insulin (decrease in proinsulin); reducing hepatocyte ER stress, inflammation, and oxidative stress; reducing hepatic fat accumulation; and reducing hepatic fibrosis.

We have previously demonstrated that IL-22 is a powerful endogenous paracrine suppressor of oxidative and ER stress in pancreatic islets and that in obesity-induced hyperglycemia in mice, IL-22 therapy restored glucose control by attenuating defects in β-cells insulin

biosynthesis and secretion[5]. In addition, we have also recently demonstrated that endogenous pancreatic β-cell IL-22RA1 signalling plays a crucial role in modulating β-cell health and function[22]. Whilst our data suggests that the major mechanism by which IL-22 restores glycemic control in Type-2 Diabetes is by direct effect on the pancreatic islets, IL-22RA1 is also expressed on mucosal epithelial cells, epidermal cells, and hepatocytes[8]. Here we show that IL-22 has multimodal effects on the liver and is therefore a promising therapeutic for use in MASH. Based on experimental IL-22 transgenic animals, IL-22RA1 expression patterns and known roles for IL-22 in inflammation and wound repair, systemic IL-22 has the potential to cause adverse effects, and this has been borne out in clinical trials of IL-22Fc. We confirmed that long-term systemic administration of long-acting versions of IL-22 like IL-22-Fc, can lead to hyperproliferation, specifically in skin and intestine with a high cellular turnover rate. In addition to increased proliferation, an increase in inflammation was also noted. Therefore, our goal was to harness the therapeutic efficacy of IL-22, by designing prototype fusions of IL-22 to enhance targeting to the liver and the pancreas. Two parallel approaches were taken to design the targeted IL-22; fusions with GLP1 receptor binding proteins or fusions with single-chain antibody domains that were reactive with pancreatic and hepatic antigens.

The fusion with the single-chain antibody domain (IL-22-ScFv) does not target IL-22RA1 in the intestine and skin, whilst enhanced targeting is observed in the liver and pancreas. By designing variations of the linker that separate the functional domains (IL-22 and ScFv) we chose the lead fusion based on the targeting and efficacy observed. Moreover, compared with native IL-22, IL-22-ScFv was more stable and able to potently activate STAT3p even after 7 days storage at room temperature. Importantly, IL-22-ScFv targeting reduced intestinal Stat3p to negligible levels, whilst a 50-100-fold intestinal Stat3 activation was observed with native IL-22 and IL-22-Fc. Some epidermal Stat3p activation at the injection site was observed with IL-22-ScFv treatment, which was to be expected. The putative risks of chronic IL-22 therapy are predominantly a risk of hyperproliferation or tumor development in the intestine or skin. However, we did not observe any increase in intestinal villus length and epidermal thickness with IL-22-ScFv, as was observed with IL-22-Fc. Therefore, this approach of targeting IL-22, appears to effectively diminish the exposure in the intestine and skin; greatly reducing the risks associated with hyperproliferation.

In obese mice native IL-22 administration has multiple highly desirable physiological and therapeutic effects including decreased body weight and a decrease in lipid accumulation in fat depots. IL-22 is known to activate transcription factors in primary adipocytes[23] and increase expression of genes involved in triglyceride lipolysis, which may explain the changes in adipocyte size reported[8] and as we noted with rIL-22 treatment. IL-22-driven weight loss could not be explained through an alteration in metabolic rate or energy expenditure. Whilst reduced food consumption could account for these alterations, it could not explain the improvements in hyperglycemia in this study[8]. A significantly lower cumulative food intake in obese mice over the 12 days of biweekly treatment with IL-22 was observed, which was accompanied by the expression of hypothalamic genes known to enhance satiety. We did not observe weight loss with the targeted IL-22 fusions when administered subcutaneously, despite the improvements in the insulin quality. Therefore, the improvements observed with targeted IL-22 are independent of any effects on body weight.

We observed a significant reduction in hepatic steatosis with IL-22-ScFv across multiple in-vivo models and a marked decrease in lipid-containing HEPG2 cells. Our proteomics analysis demonstrates IL-22-ScFv drives a complex modulation of metabolic pathways associated with the reduction in lipid accumualtion, which would require much further experimental work to ascertain relative importance of individual proteins. Overexpression of IL-22 by gene targeting or delivery through adenovirus expressing IL-22 has previously been shown to reduce liver fibrosis and accelerate fibrosis resolution during recovery in the carbon tetrachloride ($CCl_4$) model[19,24]. We used multiple independent preclinical MASH models to assess the effect of IL-22-ScFv, as all models reflect different aspects of human MASH[25]. Using the CDAHFD diet-based model, where animals develop fatty livers with fibrosis, IL-22-ScFv reduced NAS scores and hepatic fibrosis, accompanied by decline in gene expression of key fibrosis genes. A decrease in bridging fibrosis was seen in the TAA model, when combined with an HFD. This was attributed to IL-22-driving senescence in hepatic stellate cells directly through STAT3 and SOCS3 activation, corroborating previous reports[19,24]. In-vitro IL-22-ScFv potently reduced expression of fibrosis markers and enhanced the expression of *SOCS3* in stellate cells. In all the models utilized (HFD, HFD/TAA and CDAHFD), a substantial decrease in hepatic inflammation was apparent compared with IL-22-Fc. This could be driven by the impact of IL-22-ScFv on ER and oxidative stress and increase in *SOCS3*, which is an inducible negative regulator of JAK/STAT pathway.

We demonstrate that IL-22-ScFv works via a direct receptor-mediated effect, improving multiple pathways activated in MASH. Targeting more than one pathway in MASH -pathology is important in treating a broader range of MASH patients, from early stage through to more advanced conditions, and with differing underlying complex aetiologies. IL-22-ScFv modulated all four pathways in MASH that are currently being targeted in clinical trials: (1) hepatic fat accumulation; (2) hepatic inflammation; (3) upstream metabolic changes (glucose tolerance, insulin tolerance); and (4) fibrosis. Targeting IL-22 enhances the protective effects in the pancreas and liver, without causing off-target effects, such as hyperproliferation, in the skin and gut. Targeted IL-22 therapy therefore has the potential to have a profound effect on MASH pathology.

## Methods
### Animal experiments
All mice were housed in sterilized, filter-topped cages in a conventional clean facility. Animals were maintained on a 12:12-h light-dark cycle, at ambient temperature (22 - 23 °C), 40-60% humidity and received food (specific diets specified below) and water *ad libitum* with nesting materials. All experiments were approved by the University of Queensland Animal Ethics Committee (Ethics #2021/AE000426, AE519/16) and conducted in accordance with guidelines set out by the National Health and Medical Research Council of Australia. Detailed treatment regimens are shown in the figures. Of note, the cages of mice were randomly allocated to the experimental groups. Investigators were not blind to treatment, but no subjective assessments were made; histological assessment was conducted blind. At the conclusion of each experiment, animals were euthanised via cervical dislocation, with the absence of hind foot pinch response confirming successful euthanasia.

**HFD Experiments.** 6-8 weeks old *C57BL/6* male mice were fed *ad libitum* a lard high-fat diet (HFD-lard; Speciality feeds, SF04-001) or normal chow diet (NCD; Speciality feeds, SF00-100) containing less than 10% saturated fat. At the end of the experiment, mice were fasted for 5 h before sacrifice.

**HFD-TAA Experiments.** 6-8 weeks old C57BL/6 male mice were fed ad libitum a lard high fat diet (HFD-lard; Speciality feeds, SF04-001) or normal chow diet (NCD; Speciality feeds, SF00-100) containing less than 10% saturated fat for 16 weeks. Animals in the HFD-TAA group remained on HFD, with the addition of 300 mg/L Thioacetamide (TAA; Sigma-Aldrich 163678) in their drinking water, for a further 12 weeks. All experimental animals were treated biweekly, during the last 4 weeks of the experiment.

**CDAHFD Experiments.** 6-8 weeks old C57BL/6 male mice were fed ad libitum a low choline low methionine 25% fat diet (CDAHFD; Speciality feeds, SF05-016) or a normal chow diet (NCD; Speciality feeds, SF00-100) containing less than 10% saturated fat for 11 weeks. All experimental animals were treated biweekly, during the last 5 weeks of the experiment.

**IL-22ra1[β-cell−/−] HFD Experiments.** *Il22ra1[fl]* mice (strain #031003) and Ins2-cre mice (strain #003573) were purchased from the Jackson Laboratory and crossed to produce Il22ra1 x Ins2-cre animals. 8-week-old male mice were fed ad libitum a lard high fat diet (HFD-lard; Speciality feeds, SF04-001) for 12 weeks. A subset of these animals received biweekly treatment, during the last 2 weeks of the experiment.

**IL-22ra1[Hep−/−] HFD Experiments.** *Il22ra1[fl]* mice (strain #031003) and Alb-cre mice (strain #003574) were purchased from the Jackson Laboratory and crossed to produce Il22ra1 x Alb-cre animals. 8-week-old male mice were fed ad libitum a lard high fat diet (HFD-lard; Speciality feeds, SF04-001) for 12 weeks. A subset of these animals received biweekly treatment, during the last 2 weeks of the experiment.

### Production of IL-22-fusion proteins

**Cloning.** Sequences for human IL-22 and the pancreas targeting single chain antibody fragment (ScFv) were codon optimized for heterologous expression in Chinese hamster ovary (CHO) cells and synthesized by GeneArt (Thermo Fischer Scientific). The complete sequence for the immunocytokines (Supplementary Fig. 4a) was assembled via Gibson assembly (New England Biolabs). The construct was cloned into the mammalian expression vector pcDNA3.1(+) (Invitrogen) harboring a Kozak sequence and a human IgG2 heavy chain signal sequence (MGWSCIILFLVATATGVHS) via 5′ XhoI and 3′ XbaI restriction sites.

A XhoI restriction site and c-myc tag as well as a His$_6$ tag and XbaI restriction site were attached via PCR to the codon optimized sequence of human IL22 at the 5′ and 3′ end, respectively. The resulting construct was cloned into the same pcDNA3.1(+) backbone.

**Expression.** CHO-S cells (Invitrogen) were grown in humidified atmosphere supplemented with 7.5% CO$_2$ at 37 °C, 125 rpm in CD-CHO medium (Gibco) and passaged for at least 5 times prior to transfection. Cells were kept at >95% viability for all transfection experiments. At the day of transfection, cells were collected by centrifugation (300xg, 5 min) and resuspended in fresh CD-CHO medium at a density of 3×10$^6$ cells/ml. 2 ug of DNA/ml initial culture volume was complexed with the transient transfection agent PEI MAX (Polyscience Inc.) at a ratio of 1:4 (w:v) using OptiPro (Gibco) as complexing medium. DNA was complexed for 15 min on room temperature before being added to the cell culture. Cells were allowed to grow for 5 h at 37 °C, 125 rpm, 7.5% CO$_2$ to facilitate uptake of DNA. After 5 h, the cells were fed with twice the initial transfection volume CD-CHO supplemented with 12% (v/v) Efficient Feed A (Gibco) 12% (v/v) Efficient Feed B (Gibco) and 0.6% (v/v) anti-clumping agent (Gibco). The temperature was decreased to 32 °C and proteins were expressed until cell viability dropped below 80% (10–12 days). Cells were harvested at 5000xg, 15 min at 4 °C and the supernatant (SN) was further cleared by passing through a 0.22 μm filter prior to purification.

**Purification.** The filtered SN containing the secreted immunocytokine was topped up 1:3 with protein L binding buffer (20 mM Tris-Cl, 150 mM NaCl, pH 7.2) and loaded on a prepacked and equilibrated protein L affinity column (GE Healthcare). The column was washed with binding buffer and the target protein was eluted in 5 column volumes (CV) elution buffer (100 mM glycine, pH 2.2). The acidified eluate was immediately neutralized to pH 7.4 using an appropriate amount of 1 M Tris-Cl, pH 8. Prior to concentration, the buffer was exchanged to PBS, pH 7.4 by means of dialysis or ultrafiltration. The protein was concentrated to 0.5–1 mg/ml using ultrafiltration devices with a molecular weight cutoff (MWCO) of 30 kDa (Amicon) for further analysis. For animal experiments, the purified protein was sterilized by filtration using a 0.22 μm filter. The average yield was around 3.5 mg pure protein per L cell culture. For hIL-22: The filtered SN containing human IL22 was directly applied on a prepacked and equilibrated HisTrap Excel column (GE Healthcare). The column was washed with binding buffer (50 mM Tris-Cl, 500 mM NaCl, 15 mM imidazole, pH 7.4) and the target protein was eluted in 10 CV elution buffer (50 mM Tris-Cl, 500 mM NaCl, 500 mM imidazole, pH 7.4). Fractions containing the target protein were pooled and extensively dialyzed against anion exchange binding buffer (20 mM Tris-Cl, 50 mM NaCl, pH 7.6). The dialyzed solution was applied on a prepacked and equilibrated HiTrap Q anion exchange column (GE Healthcare) and the flow-through (FT) containing the pure target protein was collected. Buffer exchange and protein concentration was achieved using an ultrafiltration device with a MWCO of 10 kDa. For animal experiments, the purified protein was sterilized by filtration using a 0.22 μm filter. The average yield was around 7 mg pure protein per L cell culture.

**Analytical size exclusion chromatography (HPLC-SEC).** For analytical purposes, 50 μL of purified proteins with a concentration of 0.5 mg/mL were applied on a pre-packed TSKgel 3000SWXL column. The column was run with a constant flow rate of 0.8 mL/min with PBS + 200 mM NaCl (pH 7.3) as the mobile phase. A gel filtration standard (670/158/44/17/1.35 kDa) was used to calculate the apparent molecular weight of the eluting fractions.

### Metabolic measurements

For oral glucose tolerance test (oGTT), mice were fasted for 5 h and then challenged with 50 mg of D-Glucose solution by oral gavage. Blood glucose levels were measured using a glucometer (SensoCard, 77 Electronika, Hungary) via tail bleeding at 0, 15, 30, 60, 120 min time-points post-challenge. For i.p. insulin tolerance tests (ipITT), mice were fasted for 5 h, and then challenged with 0.75 U per kg body weight of insulin (Humalog, Lilly). Blood glucose levels were then measured using glucometer via tail bleeding at 0, 15, 30, 60, 120 min time-points post-challenge.

### qRT-PCR

Tissue samples were snap-frozen at the time of harvest for RNA isolation. Tissue was directly homogenized in Trizol (Thermo Fisher Scientific) using FastPrep tissue homogenization system (MP Biomedical). RNA was isolated using the standard phenol-chloroform method. RNA was extracted from cultured cells using RNA extraction kit (isolate II RNA mini kit, Bioline) according to the manufacturer's instructions. cDNA was then synthesized using SensiFAST (Bioline) cDNA synthesis kit containing oligo (dT) and random hexamers. Equal amount of RNA was used for all the samples. Gene expression was then measured on Viia7 Real-Time PCR System (Life Technologies) using Low ROX SYBR Green (Bioline). Primer sequences are included in Supplementary Table 1, and efficiencies were determined using cDNA dilutions and primer dilutions for the genes of interest. All data are normalized against housekeeping genes (as indicated in the figure legends, primer sequences shown in Supplementary table 2) and expressed as a fold difference to the mean of relevant control group samples.

### Human islets

We obtained approval for procuring and performing experiments with human islets from the Mater Health Services Human Research Ethics Committee. We obtained human pancreatic islets from 4 organ donors via the Tom Mandel Islet Transplant Program in Australia. Human islets

were picked under a dissection microscope and rested overnight. Islets were treated with 10 μM of Tunicamycin for 6 h, washed and isolated for gene expression and protein analyses 18 h later. Total proinsulin content was measured in 50 islets through ELISA.

## PCR array

RT$^2$ Profiler PCR Arrays (Qiagen) for Fatty Liver (mouse) and Fibrosis (mouse) were used according to manufacturer's instructions. Briefly, cDNA was preformed using RT$^2$ First Strand Kit (Qiagen). 1 μg of cDNA was then added to RT$^2$ SYBR Green Mastermix (Qiagen) and subsequently aliquoted into the PCR Arrays. Gene expression was then measured on Viia7 Real-Time PCR System (Life Technologies) and data analysis was performed using the Qiagen GeneGlobal Data Analysis Center.

## ELISA

Insulin and proinsulin were measured using kits from Mercodia following the manufacturer's instructions. Circulating serum total and HMW adiponectin were measured using mouse HMW and Total Adiponectin ELISA (ALPCO) according to the manufacturer's protocol.

## Cell culture

**MIN6N8 cells.** MIN6N8 cells, kindly gifted by Professor Josephine Forbes, were cultured as previously described[5], in phenol-red–free Dulbecco's modified eagle's medium (DMEM; Life Technologies) containing 25 mM glucose (3.4 g L$^{-1}$ sodium bicarbonate, 50 U mL$^{-1}$ penicillin and streptomycin, 71.5 μM β-mercaptoethanol and 10% heat-inactivated fetal bovine serum (FBS).

**HEPG2 cells.** HEPG2 cells were obtained from the American Type Culture Collection (ATCC, Manassas, Virginia, USA). Cells were maintained in DMEM (Life Technologies) supplemented with 10% heat inactivated FBS, 100 U/ml penicillin and 100 mg/ml streptomycin, in a humidified atmosphere of 5% CO2 in air at 37 °C. HEPG2 cells were treated with 0-250 μM of palmitic acid to induce steatosis (Sigma-Aldrich). Solutions of palmitate-BSA complex used were prepared as previously described[5]. To assess the effects of IL-22-ScFv in steatotic hepatocytes, palmitate treated HEPG2 cells were further treated with hIL-22-ScFv for up to 72 h (with IL-22-ScFv being replenished every 24 h).

Following treatment, cells were stained with 10 μM of Bodipy$_{493/503}$ (D3922, Invitrogen) for 20 min in the dark before washing extensively with PBS. A subset of cells was incubated with primary antibodies against ACC1 (1:200; 3676; Cell Signaling Technology) for 1 h at room temperature, washed extensively with PBS, and further incubated with Alexa Fluor 647 Goat anti-rabbit IgG (1:500; A32733, Invitrogen) for 1 h at room temperature. Following extensive washing with PBS, ACC1 and Bodipy staining were determined using a PHERAstar plate reader. Representative images of Bodipy staining were captured using the Olympus confocal microscope FV3000.

For mass spec analyses: Treated and control cell pellets were lysed by resuspension in 100 mM Tris pH 8.5, 1% Sodium Deoxycholate, 40 mM 2-chloroacetamide, 10 mM Tris (2-carboxyethyl) phosphine. Lysates were disrupted by sonication on a Bioruptor and heat inactivation (90$^\circ$C for 5 min). 20 μg of each lysate sample was digested with trypsin according to the manufacturer's instructions (Promega catalog V5111). Digests were acidified and peptides desalted using C18 STAGE tips according to the manufacturer's instructions (Agilent catalog A57003100). Tryptic peptides resolved by C-18 separation on an UltiMate 3000 HPLC nano-flow HPLC and analyzed on Q Exactive Plus orbitrap mass spectrometer running in positive ion data-dependent analysis (DDA) mode, with settings typical of peptide analyses. Raw data was searched against the human proteome using the Proteome Discoverer 3.0 platform and the Chimerys search engine. Peptides were quantified by precursor abundance,

and group comparisons were made using pairwise peptide ratios and a background-based t-test.

**LX-2 cells.** LX-2 cells, kindly gifted by Professor Grant Ramm, were maintained in DMEM (Life Technologies) supplemented with 2% heat-inactivated FBS, 100 U/ml penicillin and 100 mg/ml streptomycin in a humidified atmosphere of 5% CO2 in air at 37 °C. Prior to experiments, cells were serum starved in media supplemented with 0.1% FBS overnight.

## Immunofluorescence staining

For immunofluorescence staining, frozen tissue sections were cut at 7-8 μm, air dried and permeabilized with ice-cold methanol for 15 min. Sections were briefly washed twice with PBS and placed in 0.3% Triton X-100 10 min before blocking with 10% KPL (5140-0011, SeraCare) in PBS for 1 h at room temperature. The sections were incubated with primary antibodies against phospho-Stat3 (1:200; 9145, Cell Signaling Technology), insulin (1:200; PA1-26938, ThermoFisher), proinsulin (1:200; MAB-13361, R&D), Reelin (1:1000; AF3820, R&D), and GFAP (1:200; LS-B4775, LifeSpan Bioscience) at 4 °C overnight. Sections were washed three times with PBS and incubated for 1 h in the dark with the corresponding fluorophore-conjugated secondary antibodies, Alexa Fluor 488 Chicken anti-rabbit IgG (1:500; A-21441, Invitrogen), Alexa Fluor 488 Goat anti-guinea pig IgG (1:1000; A-11073, Invitrogen), Alexa Fluor 555 Goat anti-rabbit IgG (1:1000; A-32732, Invitrogen), Alexa Fluor 555 Goat anti-chicken IgG (1:500; A-21437, Invitrogen), Alexa Fluor 647 Goat anti-mouse IgG (1:500; A-21235, Invitrogen). Sections were counterstained using DAPI (1:1000 from 1 mg/ml; D9542; Sigma) to visualize cell nuclei and cover slipped with Fluoroshield (Sigma, F6182). Images were captured using the Olympus confocal microscope FV3000. ImageJ software was used to quantify MFI per islet area in the Il22ra1 x Ins2-cre animals.

## Immunohistochemistry staining

Paraffin-embedded tissue sections were dewaxed and underwent antigen retrieval via sodium citrate heating for 25 min. Endogenous peroxidase was blocked using 3% H$_2$O$_2$ in PBS, before being blocked with 10% KPL (5140-0011, SeraCare) in PBS for 1 h at room temperature. The sections were incubated with primary antibodies against Ki67 (1:1000; MA5-14520, ThermoFisher Scientific) at 4 °C overnight. Sections were washed three times with PBS and incubated with anti-rabbit IgG (HRP) secondary antibody (1:200; 32260, Invitrogen) for 1 h at room temperature. Immunohistochemical staining was observed using 3,3'-diaminobenzidine (DAB) substrate kit (K3468, DAKO) and was counterstained with hematoxylin.

## TUNEL assay

In situ nick end-labeling (TUNEL) assay was performed using the in situ cell death detection kit, Fluorescein (Roche, Applied Science. Cat. No.11684795910) as per manufacturer's instructions. Sections were counterstained using DAPI (1:1000 from 1 mg/ml; D9542; Sigma) to visualize cell nuclei and cover slipped with Fluoroshield (Sigma, F6182). Images were captured using the Olympus confocal microscope FV3000.

## Quantification of lipid contents

Oil Red O (ORO) (2 g; Sigma, O-0625) was dissolved in 200 ml of 100% Propan-2-ol; swirled to mix and left overnight at room temperature. A working solution of ORO was made with 6 ml of ORO stock solution and 4 ml of distilled water. The working solution was filtered using Whatman #1 filter paper, left to stand for 30 min, and filtered (Whatman #1) again before use. Cut frozen sections at 7-8μm were fixed with 10% formalin for 10 min at room temperature and washed briefly with running water. The sections were stained with freshly prepared ORO working solution for 15 min at room temperature after rinsed with 60%

isopropanol for 30 s. The sections were rinsed with 60% isopropanol for 5 s. Next, the sections were rinsed with running tap water for approximately 1 min followed by immersion in Mayer's hematoxylin solution (Sigma, MHS-80) for 5 min. The sections were washed with running tap water, then mounted with Fluoroshield (Sigma, F6182) an aqueous-based preservative. The lipid contents from the stained sections were quantified using Visiopharm software. Lipid droplet size was measured using the ImageJ software.

## Histological analysis

Adipocyte size was measured using ImageJ software. The thickness of the injected and uninjected epidermis layer from H&E staining was measured 3 times at different areas for each image using Image J software. The average of the three measurements of each sample was used for the statistical analysis. Liver pathology was scored using the NASH CRN system blinded: (NAS score; sum of 0–8), as a composite of steatosis (0–3), Hepatocyte ballooning (0–2), and lobular inflammation (0–3). Fibrosis staging was scored (0–4) with 0 = no fibrosis; 1a = zone 3 mild perisinusoidal fibrosis, 1b = zone 3 moderate perisinusoidal fibrosis, 1c = portal or periportal fibrosis only; 2 = zone 3 and portal/periportal fibrosis; 3 = bridging fibrosis; and 4 = cirrhosis.

## Western blots

Cells were rinsed twice, with cold PBS, and lysed in ice-cold RIPA buffer (150 mM sodium chloride, 1.0% Triton X-100, 0.5% sodium deoxycholate, 0.1% SDS, 50 mM Tris, pH 8.0; Thermo Fisher Scientific), supplemented with complete Protease Inhibitor Cocktail (Sigma Aldrich) and PhosStop (Thermo Fisher Scientific). Protein concentrations were determined using Pierce BCA Protein Assay Kit (Thermo Fisher Scientific) as per the manufacturer's protocol. Up to 30 μg of protein per sample was loaded on NuPAGE 8–12% Bis-Tris protein gels (Thermo Fisher Scientific) and transferred onto iBlot2 Transfer Stacks PVDF membrane (Thermo Fisher Scientific). Membranes were blocked with Odyssey Blocking buffer (Milennium Science, Victoria, Australia) for 1 h, probed with rabbit anti-STAT3p antibody (1:200; 9145, Cell Signaling Technology) and mouse anti-β-actin (1:5000; NB600-501, Novus Biologicals) overnight, followed by detection with goat anti-rabbit IgG DyLight 800 (1:20,000; Thermo Fisher Scientific) and goat anti-mouse IgG DyLight 600 (1:20,000; Thermo Fisher Scientific). Bands were visualized using the Odyssey CLx system (LI-COR Biosciences, Nebraska, USA) and analyzed with Image Studio Lite V5.2 (LI-COR Biosciences).

## Statistical analysis

The in vivo experiments were powered for a 1.5 s.d. change in the area under the curve of the glucose tolerance test as the major primary outcome measure. Statistical analyses were performed using GraphPad PRISM version 9.2.0 (GraphPad software, Inc.) as described in individual figure legends. After confirmation of a normal distribution by probability plots, differences between groups were assessed by using parametric tests (one-way ANOVA with post-test or, where appropriate, a two-tailed Student t-test). Where a normal distribution could not be confirmed, the Kruskal-Wallis non-parametric ANOVA with a Dunn's post hoc test was used.

## Reporting summary

Further information on research design is available in the Nature Portfolio Reporting Summary linked to this article.

## Data availability

The proteomics dataset generated and analyzed during the current study is available on the ProteomeXchange repository, under accession code PXD051262. All other data generated in this study are provided in the Supplementary Information/Source Data file. Source data are provided with this paper.

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

## Acknowledgements

The authors would like to thank the staff of the Mater Research and Translational Research Institute Biological Research Facilities for care of breeding and experimental animals. The authors thank Thomas Loudovaris, Thomas W Kay and Helen E Thomas (St. Vincent's Research Institute, Melbourne, Victoria, Australia) for providing the human pancreatic islet samples. Thank you to the Protein Expression Facility (PEF, The University of Queensland) for the recombinant protein production and National Biologics Facility, UQ for the protein analytics. A special thank you to A/Prof Katharine Irvine (Mater Research Institute-University of Queensland) for her advice regarding the long-term fibrosis models. This work was supported by Australian National Health and Medical Research Council Grants (Ideas and Development to S.Z.H., J.B.P., M.A.M., G.A.R., G.A.M., R.T.B.) and Fellowships (Career Development Fellowship-1 to S.Z.H.), Gastroenterological Society for Australasia (Clinical Collaboration Grant to S.Z.H. and G.A.M., Lawrie Powell Award to S.Z.H. and Project Grant to S.Z.H.), Mater Foundation (to S.Z.H.) and Jetra Therapeutics (S.Z.H., J.B.P., and M.A.M.).

## Author contributions

S.Z.H., M.A.M. and J.B.P. conceived the study. H.S conducted fibrosis model experiments and ex-vivo analyses of IL-22-ScFv mediated effects. S.K conducted work in the HFD models and PK studies, which were supported by A.F., R.W. K.Y.W. provided technical support for all experiments and conducted toxicity and PK experiments. D.J.B., F.J.S., T.H., and S.Z.H. conducted the phenomaster experiments and analyses. C.F., K.T., B.T., R.T.B. produced and tested the IL-22-fusions stability. A.M. provided support for all fibrosis model experiments. M.M., R.W., and Y.S. conducted in-vitro pharmacokinetic testing of IL-22-fusions. G.M. analyzed the pathology of livers., J.L. conducted microarray analyses and lipid analyses on HFD livers. V.S. provided statistical advice and supervised students. J.L.H. and D.L. conducted the mass spec experiment and analyses supported by H.S. and S.Z.H. S.Z.H., R.T.B., G.A.R., G.A.M., J.B.P., M.A.M., L.B. provided resources, funding, and intellectual input. S.Z.H. supervised the study, designed the experiments, and wrote the first draft of the manuscript. G.A.R., G.A.M., J.B.P., M.A.M., H.S., S.K., R.T.B., F.J.S., A.M., D.J.B. edited the manuscript and all authors read and approved the manuscript.

## Competing interests

S.Z.H., M.A.M., and J.B.P. are inventors on a patent relating to IL-22 use in metabolic disease. The remaining authors declare no other competing interests.

## Additional information

¹Immunopathology Group, Mater Research Institute-The University of Queensland, Translational Research Institute, Brisbane, Australia. ²Faculty of Medicine, The University of Queensland, Brisbane, QLD, Australia. ³Australian Research Council Training Centre for Biopharmaceutical Innovation, Australian Institute for Bioengineering and Nanotechnology, The University of Queensland, Brisbane, Australia. ⁴School of Chemistry and Molecular Biosciences, Faculty of Science, The University of Queensland, Brisbane, QLD, Australia. ⁵Envoi Specialist Pathologists, Kelvin Grove, Brisbane, Australia. ⁶Proteomics Core Facility, Translational Research Institute, Brisbane, Australia. ⁷Department of Respiratory and Sleep Medicine, Mater Health, South Brisbane, Australia. ⁸Department of Endocrinology & Diabetes, Queensland Children's Hospital, South Brisbane, QLD, Australia. ⁹Children's Health Research Centre, Faculty of Medicine, The University of Queensland, Brisbane, Australia. ¹⁰Department of Chemical Pathology, Mater Pathology, South Brisbane, QLD, Australia. ¹¹Department of Diabetes and Endocrinology, Princess Alexandra Hospital, Brisbane, QLD, Australia. ¹²Hepatic Fibrosis Group, QIMR Berghofer Medical Research Institute, Brisbane, QLD, Australia. ¹³Department of Gastroenterology and Hepatology, Princess Alexandra Hospital, Brisbane, QLD, Australia. ¹⁴Health Translation Queensland, Royal Brisbane and Women's Hospital, Herston, Australia. ¹⁵Faculty of Medicine, Dentistry and Health Sciences, University of Melbourne, Victoria, Australia. ¹⁶Australian Infectious Disease Research Centre, University of Queensland, Brisbane, Australia. ¹⁷These authors contributed equally: Haressh Sajiir, Sahar Keshvari. ✉e-mail: sumaira.hasnain@mater.uq.edu.au

