## [Peer Review File · Nature Communications]

Liver and Pancreatic-Targeted Interleukin-22 as a Therapeutic for Metabolic Dysfunction-Associated SteatohepatitisREVIEWER COMMENTS

Reviewer #1 (Remarks to the Author):

This study investigated a designed short-acting Interleukin-22 as a Therapeutic for Non-Alcoholic Fatty Liver Disease and Steatohepatitis. The authors demonstrated that IL-22-ScFv fusion proteins which is fusion GLP (receptor binding proteins) and single chain variable fragment (ScFv) with IL-22 increased liver and pancreas targets, eliminates the side effects. This is an interesting study. However, there are some concerns as below:

Major:

- 1 The designed short-acting IL-22-bispecific biologic drugs is novel in this study, however the introduction and advantages of this fusion protein is missing in the abstract and introduction parts.
- 2 Fig 2b, the target of IL-22-GLP1 is supposed to be pancreas, why there were no higher p-STAT3 activity in pancreas compared hIL-22 and scFv-IL-22?
- 3 In Figure3 showing the IL-22-ScFv which targets of liver and pancreas improves metabolic syndrome. Why does natural IL-22 have no effects on metabolism, whereas the IL-22-ScFv affects metabolism? Studies have suggested that rIL-22 has no effects on metabolism, and it's treatment effects in fatty liver by suppressing appetite through inhibiting the signaling of hypothalamus, so how about the effects of IL-22-ScFv on hypothalamus?
- 4 They found that IL-22 improved HFD-induced steatosis and regulated lipid metabolism gene expression (Figure.3). In addition, they also found that IL-22 improved NASH-induced fibrosis and regulated fibrotic degradation gene (Figure.4). They found that IL-22 up-regulated CPY1A, ACACA, and ChREBP mRNA expression in vitro (Figure.5). However, all protein level is not shown. They must detect protein expression in these experiments.
- 5 the authors have adopted IL-22R1bcell^{-/-} mice to improve the function of IL-22 in NAFLD and NASH dependent on pancreas b cells. To improve the liver function, we would like to suggest using Alb-IL-22R1^{-/-} mice and HSCs specific knockout mice.

Minor:

- 1 there were mentioned LX2 cells results in Figure 7 legend, but the figure is missing.
- 2 Supplementary Figure 9 is completely inconsistent with the figure legend and results.
- 3 Supplementary Figure 13d showed the activation of p-stat3 in LX2 cells is confused.
- 4 In line 529, the authors wrote that we obtained human pancreatic islets from 5 organ

donors via the Tom Mandel Islet Transplant Program in Australia, where are the results?

5 the figure legend of FIG2 g and h is incorrect.

6 the variation in figure2D is too big.

Reviewer #2 (Remarks to the Author):

Keshvari and colleagues have evaluated the effects of a short acting, liver/pancreas targeted IL22 analog on metabolic parameters in mice. While the findings might be clinically relevant and important, the studies presented are poorly described, lack important controls, and provide little mechanistic insight. These studies are judged to be very preliminary and underdeveloped.

The scope of the paper is too broad and fails to study most of the observed effects with any mechanistic detail. There is very little mechanistic data provided and what metabolic data are shown are strongly overinterpreted and fail to provide insight into what is really happening. There are no aspects of the paper that are adequately explored mechanistically and the work is mostly observational. There are also several aspects of the paper that are tangential.

The reviewer did not see any indication that the observed negative effects of long acting IL22 were not recapitulated with the use of IL22scfv.

The description of the work needs to be improved. The nature of the IL22scfv needs to be better described for readers who are not familiar with this technology. There is no description of what the control HFD mice or other groups were treated with. The lack of extensive studies using the IL22RA KO mice is also judged to be a weakness.

Several parts of the paper are tangential or understudied:

1. The Phenomaster data and body weight effects in supplemental figure 1 are confusing and underdeveloped. It is not clear how these changes in brain neuropeptides are elicited by rIL22. Were phenomaster data analyzed by ANCOVA for body weight? What's happening with fecal fat absorption? Something is missing here. Since these parameters were not

affected by use of short acting IL22, this all seems tangential and understudied.

2. Similarly, the single paragraph on the long acting IL22 seems tangential and underdeveloped.

3. The mechanism and importance of the IL22-scfv and GLP1 fusion effects on insulin and pro-insulin are unclear. There is no evidence that this direct on beta cells (or liver). How do they mediate this effect? Why doesn't GLP1 increase insulin secretion? The reviewer is unclear how this is interpreted as a beneficial readout.

4. How does the palmitate-induced ER stress readout fit into the therapeutic mechanism in vivo.

5. Why does IL22-Scfv lead to reduced fat mass?

6. The conclusion from bcell KO mice that bcell and hepatocyte IL22RA are both required is not supported by the data for multiple reasons. No hepatocyte KO mice were studied. No WT mice were included in the study design to show that the IL22-scfv elicited an effect in this study. This is sloppy science.

Several aspects of intermediary metabolism/physiology are poorly described or understood:

1. "Pck2 (Phosphoenolpyruvate carboxykinase-2) an enzyme that regulates glucose-stimulated insulin secretion" Since this is measured in liver, its role in insulin secretion is not relevant.

2. "...mRNA levels for carnitine palmitoyltransferase-1a (CPT1a) was significantly increased...Moreover, a marked increase in acetyl-coA carboxylase-alpha (ACACA) was also observed post IL-22-ScFv treatment (Fig. 5e; Supplementary Figure 13c). This suggested that IL-22-ScFv was enhancing lipolysis and inhibiting de novo fatty acid biosynthesis." Cpt1a has nothing to do with lipolysis. As an enzyme involved in DNL, an increase in Acaca would be predicted to increase DNL. Moreover, both enzymes are regulated mostly at a posttranslational level.

3. "To investigate this further, adipocyte-differentiated 3T3-L1 cells were treated with IL-22-ScFv. IL-22-ScFv reduced stored lipid content which was associated with an increase in ChREBP mRNA expression (Fig. 5f,g)." Chrebp regulates the genes that promote conversion of carbohydrates to fatty acids through de novo lipogenesis and would be associated with an increase in lipid storage.

4. Is the effect on DNL enzyme expression direct or a consequence of altered systemic

metabolism? (insulin and glucose concentrations)

5. For the GTT, challenging mice with glucose without administering it on a per gram of body weight or lean mass is a substandard approach.

Minor:

Define rIL-22 at first use

The AUC data in figure 3C do not match the curves shown. The d7 and d10 curves are not much different from the control and the D13 seems much different, but this is not reflected in the AUC data.

Reviewer #3 (Remarks to the Author):

Keshvari et al aimed to maximize the therapeutic benefits of IL-22 by creating fusion prototypes that improve its targeting towards the pancreas and liver. They pursued two strategies to design one fused with GLP1 receptor-binding proteins, and the other combined with single-chain antibody domains that interact with pancreatic and hepatic antigens. The authors have carried out extensive in vivo modelling to evaluate the potential positive and negative effects of IL22 fusion molecules in obesity-induced glucose intolerance, insulin secretion quality and liver steatosis/NAFLD. With the perspective of its clinical use, authors also tested their IL-22-ScFV candidate through subcutaneous administration and explored its pancreas-independent effects using a new beta cell-specific IL₂₂RA1 KO mice. They also discussed, and demonstrated some data, with regards to cellular and molecular mechanisms of action. Overall the study is of great interest and well written; unfortunately, some errors in the legends and in the arrangement of the figures make the reading less pleasant. There are several comments to address in this review to strengthen and clarify the messages of the paper.

Major points:

1/ IL-22 is a promising therapeutic candidate that can impact multiple organs and cell types and have complementary/synergizing effects. Notably, the IL-22-ScFv enhances targeting in

the liver and the pancreas and does not target IL-22RA1 in the intestine and the skin, minimizing the risk of adversary effects. Regarding the weight loss effect, it is unclear whether this is happening during all the different IL-22 treatments and especially whether IL-22-ScFv is able to cross the brain-blood barrier, leading to decreased food intake as shown for recombinant IL-22. Therefore, for each experiment, authors should clearly show and analyse statistically the mouse body weight changes during IL-22 injections, the body weight at sacrifice and the fat mass or fat pad weights. The effect of IL-22-ScFv in the brain should be added.

2/ Despite minimizing adverse effects, one argument for the subcutaneous administration is the absence of weight loss, which is not fully convincing in Figure S9. The schematic of S9a shows the usual 12 weeks HFD + 2 weeks of treatment, while Fig. S9b shows the body weight changes between weeks 12 and 16 and not 14. Is there any body weight difference at week 14 (when the GTT is supposed to be done)? Line 382 of the discussion also states that insulin quality was validated but it is not shown. Overall, authors should better emphasize the positive effects of this mode of injection.

3/ Figure 2h shows that hIL22 did not decrease fasting glucose but 3b shows that it significantly reduced overall GTT AUC (and most probably fasting glucose too). How do authors explain this?

4/ Pictures of STAT3-P in the pancreas mainly show labeling in acinar cells (which has the highest expression of IL22RA1 according to Tabula muris). Could authors show/quantify labeling specifically in islet beta cells?

5/ Have the β cell-specific IL22RA1 KO mice been described before? Does the deletion have an effect on insulin secretion under steady-state and during HFD? This would emphasize the physiological role of IL-22.

6/ AST/ALT and systemic metabolism (GTT/ITT) are not assessed in the NASH models (CDAHFD, TAA/HFD and HFD/MCD). These should be presented as they are in vivo measures that are closer to clinical markers that are used in humans.

7/ The main message of the paper is the effect of IL-22 on the liver. Figures 5 and S13 show the effect of IL-22 on HEPG2 in vitro, yet it feels like the experiments are limited and not fully exploited (and it seems a bit disconnected that the adipocyte experiments ended up in the main figure 5f/g). In S13a, what is the effect of IL22-ScFv on HEPG2 lipid content/inflammation? What are the levels of IL-22RA1 expression in each cell type in the liver? What is the effect of IL22-ScFv on stellate cell transcriptomic profile? Are fibrotic genes affected?

Minor points :

- Figure 1a and S1a are showing the same schematic, although with different HFD duration.
- Figure 2: pictures are too small, readers cannot see labeling.
- Authors should write mIL-22 or hIL-22 in the graphs as well for clarity.
- Figure 2: Labelling is wrong for g and h.
- Figure 2h: authors should specify that treated mice are on HFD only.
- Line 176: Text should refer to Figure 2b and c.
- Not all graphs/pictures are mentioned in the text (e.g. Fig. 3a and b).
- Figure S9: Legends are wrong.
- Line 243: Psck2 is an enzyme that regulates insulin secretion in beta cells not in liver.
- Figure 4b: why using IL22-Fc and not hIL22 as a comparison here?
- Figure 5: Not clear if mice are treated ip or subcutaneously?
- Figure 1f-l showing adverse effects should be put together with Figure S10.

We would like to thank all the reviewers for their insightful comments, which helped strengthen the manuscript significantly. We have addressed all the queries raised as indicated below and altered the manuscript to incorporate all the new data/information where possible or provided this for the reviewers below as author responses (AR) per question.

Reviewer #1:

This study investigated a designed short-acting Interleukin-22 as a Therapeutic for Non-Alcoholic Fatty Liver Disease and Steatohepatitis. The authors demonstrated that IL-22-ScFv fusion proteins which is fusion GLP (receptor binding proteins) and single chain variable fragment (ScFv) with IL-22 increased liver and pancreas targets, eliminates the side effects. This is an interesting study. However, there are some concerns as below:

Major:

1. The designed short-acting IL-22-bispecific biologic drugs is novel in this study, however the introduction and advantages of this fusion protein is missing in the abstract and introduction parts.

AR01: Thank you to the reviewer for pointing this out, we have now altered the text, in the abstract and introduction, to explicitly state the advantages – in particular the targeting of the IL-22 fusion proteins to liver and pancreas without off-target effects.

2. Fig 2b, the target of IL-22-GLP1 is supposed to be pancreas, why there were no higher p-STAT3 activity in pancreas compared hIL-22 and scFv-IL-22? Explain better. GLP-1r is found in heart/kidney/brain too. Include pSTAT3 activity in these tissues too? AR02: Yes, this is correct, as Figures 2b and 2e clearly show, IL-22-GLP1 fails to enhance pancreatic targeting, with Stat3p levels in the pancreas mirroring those of native IL-22. Furthermore, IL-22-GLP1 proved ineffective in suppressing tunicamycin-induced ER stress in human islets (Fig. 2f). Similarly, new data (Fig. 2f) reveals that it fails to improve proinsulin secretion in these islets after tunicamycin treatment. While GLP1r expression is present in the heart/kidney and brain, further investigation in these organs wasn't pursued. This decision stemmed from IL-22-GLP1 failing to enhance pancreatic targeting or reduce ER stress and its associated proinsulin secretion, which were primary objectives of our study. However, the fusion did improve liver targeting and reduce targeting towards skin and intestine. This finding, now included in the text (line 175), highlights the key differences between IL-22-ScFv and IL-22-GLP1. Ultimately, the superior performance of IL-22-ScFv in achieving our aims led us to select it as the lead compound.

3. In Figure 3 showing the IL-22-ScFv which targets of liver and pancreas improves metabolic syndrome. Why does natural IL-22 have no effects on metabolism, whereas the IL-22-ScFv affects metabolism? Studies have suggested that rIL-22 has no effects on metabolism, and it's treatment effects in fatty liver by suppressing appetite through inhibiting the signaling of hypothalamus, so how about the effects of IL-22-ScFv on hypothalamus?

AR03: We have previously demonstrated that native mouse rIL-22 can improve metabolism including reduction in body weight, improved hyperglycemia, reduction in subcutaneous fat and an increase in brown fat (Nature Medicine, 2014; PMID: 25362253). We also demonstrate in the current manuscript (Sup Fig 1) that native rIL-22 effects appetite and satiety by initiating changes in the hypothalamus. Therefore, native mouse recombinant IL-22 does change the metabolic phenotype of HFD animals.

AR04: However, in your specific question with reference to Fig 3, “Why does natural IL-22 have no effects on metabolism, whereas the IL-22-ScFv affects metabolism?” so how about the effects of IL-22-ScFv on hypothalamus?

Please note, the difference with experiments in Figure 3 is the use of native HUMAN IL-22 instead of MOUSE IL-22. At comparable doses human IL-22-ScFv is more effective than human native IL-22 in the HFD mouse model, most likely due to its targeting effect on the pancreas. To increase clarity of these differences, we have now specified throughout the manuscript whether human or mouse versions of IL-22 were being used. hIL-22-ScFv is not targeted to the brain (no Stat3p was detected in the brain), we did not observe any changes in appetite. Whilst we observed a decrease in fat mass (Supplementary Fig 6c) when IL-22-ScFv is administered intraperitoneally due to an increase in brown fat (new added to Supplementary Fig 6b, shown here). When administered subcutaneously, IL-22-ScFv did not alter body fat mass (new added to supplementary Fig 9b) or have any effects of brown fat pads (data described in line 248, data is below for the reviewer). Therefore, all the effects of hIL-22-ScFv (subcutaneous delivery) are independent of any measurable effects on body weight, appetite, or satiety.

4. They found that IL-22 improved HFD-induced steatosis and regulated lipid metabolism gene expression (Figure.3). In addition, they also found that IL-22 improved NASH-induced fibrosis and regulated fibrotic degradation gene (Figure.4). They found that IL-22 up-regulated CPY1A, ACACA, and ChREBP mRNA expression in vitro (Figure.5). However, all protein level is not shown. They must detect protein expression in these experiments.

AR05: Oil Red O and MT staining which demonstrates the changes in overall protein *in-vivo*. We have now added additional data Fig 4d, demonstrating the decrease in lipid droplet size in the CDAHFD mice treated with IL-22-ScFv. We have also now conducted additional experiments in the HEPG2 cell line (Fig 5f-h) to demonstrate a direct effect of IL-22-ScFv on lipid storage.

5. the authors have adopted IL-22R1^{bcell}^{-/-} mice to improve the function of IL-22 in NAFLD and NASH dependent on pancreas b cells. To improve the liver function, we would like to suggest using Alb-IL-22R1^{-/-} mice and HSCs specific knockout mice.

AR06: We have now added new data from IL-22-RA1 x AlbCre animals (Fig 5d, e – also shown here). As expected, mIL-22-ScFv was unable to significantly alter the lipid accumulation in the liver following a 12-week-high fat diet.

Minor:

1. there were mentioned LX2 cells results in Figure 7 legend, but the figure is missing.
2. Supplementary Figure 9 is completely inconsistent with the figure legend and results.

AR07: Thank you to the reviewer for pointing this out, these are now corrected.

3. Supplementary Figure 12d showed the activation of p-stat3 in LX2 cells is confused.

AR08: To clarify our experiment demonstrating the activation of Stat3p in LX2 post IL-22 treatment, we have now added the following text to lines 343. **“hIL-22-ScFv significantly reduced fibrosis in vivo, and hIL-22-ScFv and its variants all activated pSTAT3 in LX2 human stellate cells. This indicates that hIL-22-ScFv can directly target stellate cells, the primary cell type responsible for fibrosis development (Supplementary Figure 12d).”**

4. In line 529, the authors wrote that we obtained human pancreatic islets from 5 organ donors via the Tom Mandel Islet Transplant Program in Australia, where are the results? Potentially include schematic in Figure 2 (before 2e) to make it clearer that human islets are used.

AR09: Thank you, these results were shown in Figure 2f – we have now also added the proinsulin secretion data from the human islets to this figure and clearly labelled it with a “human islet” heading (shown in AR02 above).

5. the figure legend of FIG2 g and h is incorrect.

AR10: We apologise for this error; this is now corrected.

6. the variation in figure2E is too big.

AR11: This data is generated in an unbiased manner using automated software, which calculates the staining intensity as pixels, this would include any background staining as well, which can vary. However, we have also provided the IF images (Fig 2d) so that the readers can clearly visualise that there is negligible staining in the intestine with our IL-22-targeted biologics.

Reviewer #2 (Remarks to the Author):

Keshvari and colleagues have evaluated the effects of a short acting, liver/pancreas targeted IL22 analog on metabolic parameters in mice. While the findings might be clinically relevant and important, the studies presented are poorly described, lack important controls, and provide little mechanistic insight. These studies are judged to be very preliminary and underdeveloped. The scope of the paper is too broad and fails to study most of the observed effects with any mechanistic detail. There is very little mechanistic data provided and what metabolic data are shown are strongly overinterpreted and fail to provide insight into what is really happening. There are no aspects of the paper that are adequately explored mechanistically, and the work is mostly observational. There are also several aspects of the paper that are tangential. The reviewer did not see any indication that the observed negative effects of long acting IL22 were not recapitulated with the use of IL22scfv. The description of the work needs to be improved. The nature of the IL22scfv needs to be better described for readers who are not familiar with this technology (to improve). There is no description of what the control HFD mice or other groups were treated with (to include). The lack of extensive studies using the IL22RA KO mice is also judged to be a weakness. Several parts of the paper are tangential or understudied:

1. The Phenomaster data and body weight effects in supplemental figure 1 are confusing and underdeveloped. It is not clear how these changes in brain neuropeptides are elicited by rIL22. Were phenomaster data analyzed by ANCOVA for body weight? What's happening with fecal fat absorption? Something is missing here. Since these parameters were not affected by use of short acting IL22, this all seems tangential and understudied.

AR12: We are frequently asked about the IL-22-driven changes in metabolism (please also see AR03). This is because several studies have demonstrated that native IL-22 reduces body weight (PMID: 25119041, 25362253). We have also published the extensive intestinal effects of IL-22 in the high fat diet model previously (PMID: 27350069).

In this study, we confirm the changes in fat mass with native IL-22 and demonstrate that the changes in appetite via the hypothalamic neuropeptide changes are probably independent of hyperglycemia and insulin. However, the reviewer is correct, we do not observe these changes with IL-22-ScFv, because it is targeted to the pancreas and liver and therefore this is an important point of differentiation that is highlighted by the inclusion of this data.

2. Similarly, the single paragraph on the long acting IL22 seems tangential and underdeveloped.

AR13: We have now combined these two sections into a single results section "*Long-term systemic IL-22 treatment reduced hepatic lipid accumulation but induced skin and intestinal inflammation and hyperproliferation*". Mainly to highlight that whilst native and long-acting IL-22 reduce liver fat accumulation, there are adverse effects in the intestine and skin associated with hyperproliferation. Of note, this data (Fig1f-i) is also complemented by what we show in Supplementary figure 10, demonstrating that compared to targeted IL-22 (IL-22-ScFv), IL-22-Fc resulted in increased skin proliferation, and inflammation.

3. The mechanism and importance of the IL22-scfv and GLP1 fusion effects on insulin and pro-insulin are unclear. There is no evidence that this direct on beta cells (or liver). How do they mediate this effect? Why doesn't GLP1 increase insulin secretion? The reviewer is unclear how this is interpreted as a beneficial readout.

AR14: We have shown that IL-22 can suppress ER stress, and lead to an improvement in proinsulin to insulin ratio which is a measure of beta dysfunction (Nature Med, 2014; PMID: 25362253). Here, for a head-to-head comparison of our targeted IL-22-ScFv vs native IL-22 we show that these DIRECTLY reduce beta-cell stress using the human islets (Fig 2f) and we have now added complementary data from the same experiment, demonstrating that this is accompanied by a decrease in proinsulin secretion. However, IL-22-GLP1 was not as effective as IL-22-ScFv at the same concentration. This could be due to several reasons, including the structure of the protein, which is why we chose IL-22-ScFv as our lead compound. Please also refer to AR02.

4. How does the palmitate-induced ER stress readout fit into the therapeutic mechanism in vivo. AR15: We have previously demonstrated that in obesity and metabolic syndrome, increased circulating levels of fatty acids can directly induce cellular stress in the pancreatic beta cells, resulting in insulin misfolding (PMID: 25362253). We also demonstrated that IL-22 can inhibit cellular stress initiated by free fatty acids (such as palmitate; PMID: 27350069).

5. Why does IL22-Scfv lead to reduced fat mass?

AR16: IL-22-ScFv when administered **intraperitoneally** reduced fat mass in high fat diet animals, data demonstrates that IL-22-ScFv (i.p injection) increases brown fat mass which may contribute to the reduced body fat. This is now added to Figure S6b. However, no changes are in fat mass are observed in animals when IL-22-ScFv is administered through **subcutaneous** injections, this data is now added to Fig S9b. Please also refer to AR04.

6. The conclusion from bcell KO mice that bcell and hepatocyte IL22RA are both required is not supported by the data for multiple reasons. No hepatocyte KO mice were studied.

AR17: Please also see AR06. We have now added AlbCre mice to demonstrate that hepatic IL-22RA1 is required for the improvements in lipid accumulation.

No WT mice were included in the study design to show that the IL22-scfv elicited an effect in this study. This is sloppy science.

For simplicity, we only show the data of knockout animals within this experiment as Figure 1-Supplementary 13 also demonstrate the efficacy of IL-22-ScFv in wild-type animals. These are added below for the reviewer, and we have mentioned it now in the text but have not added these to the figure.

Several aspects of intermediary metabolism/physiology are poorly described or understood:
1. “Pck2 (Phosphoenolpyruvate carboxykinase-2) an enzyme that regulates glucose-stimulated insulin secretion” Since this is measured in liver, its role in insulin secretion is not relevant.

AR18: We have now changed the wording to highlight the role of *Pck2* in the liver, as a critical player in gluconeogenesis.

2. “...mRNA levels for carnitine palmitoyltransferase-1a (CPT1a) was significantly increased...Moreover, a marked increase in acetyl-coA carboxylase-alpha (ACACA) was also observed post IL-22-ScFv treatment (Fig. 5e; Supplementary Figure 13c). This suggested that IL-22-ScFv was enhancing lipolysis and inhibiting de novo fatty acid biosynthesis.” Cpt1a has nothing to do with lipolysis (change to fatty acid oxidation). As an enzyme involved in DNL, an increase in *Acaca* would be predicted to increase DNL. Moreover, both enzymes are regulated mostly at a posttranslational level.

AR19: We agree that post-translational level regulation is important for ACACA and CPT1a. There is data demonstrating that increase in absolute CPT1a levels (through liver CPTA1a gene therapy: <https://faseb.onlinelibrary.wiley.com/doi/full/10.1096/fj.202000678R>) reduces steatosis in HFD model. Additionally, genetic inhibition of ACACA leads to an increase in fat storage (<https://www.sciencedirect.com/science/article/pii/S2212877814000453>). We have now added this reference to line 323. We have now added data for ACC1 protein, which shows a decrease with hIL-22-ScFv treatment in HEPG2-palmitate cells and this is reflected by a decrease in total lipid accumulation (determined by Bodipy staining). Please refer to AR05 for the data.

3. “To investigate this further, adipocyte-differentiated 3T3-L1 cells were treated with IL-22-ScFv. IL-22-ScFv reduced stored lipid content which was associated with an increase in ChREBP mRNA expression (Fig. 5f,g).” Chrebp regulates the genes that promote conversion of carbohydrates to fatty acids through de novo lipogenesis (DNL) and would be associated with an increase in lipid storage. 4. Is the effect on DNL enzyme expression direct or a consequence of altered systemic metabolism? (insulin and glucose concentrations).

AR20: Whilst the canonical Chrebp pathway is known to induce de novo lipogenesis, it is also known that overexpression of Chrebp can reduce concentrations of VLDL-triglycerides and low-density lipoprotein TGs. This increase in Chrebp induced by IL-22-ScFv, is reflected by the decrease in bodipy staining that demonstrates the decrease in overall lipid protein. We have now referred to these studies in the text to improve clarity (PMID: 32083759). Changes in the metabolic pathways in-vivo likely to be driven by direct and in-direct changes in metabolism, as the reviewer can appreciate this is hard to dissect in-vivo.

5. For the GTT, challenging mice with glucose without administering it on a per gram of body weight or lean mass is a substandard approach.

AR21: While the IP-GTT approach using a per gram of body weight approach has been valuable in our previous studies (PMID: 28779211; PMID: 25362253), we believe that conducting the oral glucose tolerance test (OGTT) based on a standardized glucose concentration better reflects the physiologically relevant clinical setting. This is because the incretin response, a hormonal response to oral glucose ingestion, plays a crucial role in glucose regulation and is not adequately captured by the IP-GTT method.

To align our study with the clinical setting, where glucose is measured after a 75 g oral glucose load, we administered 50 mg of glucose to mice and measured their blood glucose levels. This standardized approach allows for a more direct comparison of our findings to clinical data and provides a more accurate assessment of IL-22's effects on glucose tolerance.

Minor:

1. Define rIL-22 at first use

AR22: Thank you, we have now changed this throughout the manuscript and highlighted the use of mouse vs human versions of IL-22.

2. The AUC data in figure 3C do not match the curves shown. The d7 and d10 curves are not much different from the control and the D13 seems much different, but this is not reflected in the AUC data. AR23: Glucose tolerance test (GTT) line graphs clearly depict mean blood glucose levels over time, accompanied by error bars representing standard error of the mean (SEM). This visualization effectively showcases differences in glycemic response between treatment groups which would be lost if we present it as STDEV. Therefore, area under the curve (AUC) calculations capture the overall glycemic response for each individual animal, considering the natural fluctuations in blood glucose levels experienced within each group. This combined approach ensures both accurate representation of group trends and comprehensive analysis of individual variations in glycemic control.

Reviewer #3 (Remarks to the Author):

Keshvari et al aimed to maximize the therapeutic benefits of IL-22 by creating fusion prototypes that improve its targeting towards the pancreas and liver. They pursued two strategies to design one fused with GLP1 receptor-binding proteins, and the other combined with single-chain antibody domains that interact with pancreatic and hepatic antigens. The authors have carried out extensive in vivo modelling to evaluate the potential positive and negative effects of IL22 fusion molecules in obesity-induced glucose intolerance, insulin secretion quality and liver steatosis/NAFLD. With the perspective of its clinical use, authors also tested their IL-22-ScFV candidate through subcutaneous administration and explored its pancreas-independent effects using a new beta cell-specific IL_22RA1 KO mice. They also discussed, and demonstrated some data, with regards to cellular and molecular mechanisms of action. Overall the study is of great interest and well written; unfortunately, some errors in the legends and in the arrangement of the figures make the reading less pleasant. There are several comments to address in this review to strengthen and clarify the messages of the paper.

Major points:

1. IL-22 is a promising therapeutic candidate that can impact multiple organs and cell types and have complementary/synergizing effects. Notably, the IL-22-ScFv enhances targeting in the liver and the pancreas and does not target IL-22RA1 in the intestine and the skin, minimizing the risk of adversary effects. Regarding the weight loss effect, it is unclear whether this is happening during all the different IL-22 treatments and especially whether IL-22-ScFv is able to cross the brain-blood barrier, leading to decreased food intake as shown for recombinant IL-22. Therefore, for each experiment, authors should clearly show and analyse statistically the mouse body weight changes during IL-22 injections, the body weight at sacrifice and the fat mass or fat pad weights. The effect of IL-22-ScFv in the brain should be added.

AR24: Thank you, please also refer to AR03 and AR04. Whilst native IL-22 drives a loss in body weight, this effect is lost with IL-22-ScFv (when administered subcutaneously). We do not observe any differences in food intake with IL-22-ScFv which are clearly observed with native IL-22. This is a point of differentiation, as all the metabolic improvements with IL-22-ScFv (s.c) are independent of changes in body weight.

2. Despite minimizing adverse effects, one argument for the subcutaneous administration is the absence of weight loss, which is not fully convincing in Figure S9. The schematic of S9a shows the usual 12 weeks HFD + 2 weeks of treatment, while Fig. S9b shows the body weight changes between weeks 12 and 16 and not 14. Is there any body weight difference at week 14 (when the GTT is supposed to be done)? Line 382 (406) of the discussion also states that insulin quality was validated but it is not shown. Overall, authors should better emphasize the positive effects of this mode of injection.

AR25: We have now corrected the schematic to demonstrate that this is in fact a 2-week treatment regimen as stated in the text and corrected the graph. We have also now added the data from NMR demonstrating that no changes were observed in the fat mass in these animals (please also refer to AR04). We have also added the proinsulin: insulin ratio data here that demonstrates efficacy of the targeted fusion, compared with IL-22-Fc to Figure S10g.

3. Figure 2h shows that hIL22 did not decrease fasting glucose but 3b shows that it significantly reduced overall GTT AUC (and most probably fasting glucose too). How do authors explain this?

AR26: Improved glucose tolerance potentially precedes improvements in fasting glucose, we have demonstrated that with recombinant mouse IL-22 random fed blood glucose is decreased (Nature Medicine 2014; PMID: 25362253). Our data suggests that a longer treatment with human-IL-22 could lead to improvements in fasting glucose. We have now added this as a discussion point in the results section, line 218.

4. Pictures of STAT3-P in the pancreas mainly show labeling in acinar cells (which has the highest expression of IL22RA1 according to Tabula muris). Could authors show/quantify labeling specifically in islet beta cells?

AR26: Thank you for this comment. For the purposes of this work, we have not specifically looked at islets, as our targeting approach increases delivery overall to the pancreas, which does induce changes in the islets molecularly as well as altering the protein levels of

Proinsulin:Insulin. We also demonstrate that IL-22RA1-beta cell signaling is essential for the improvements driven by IL-22-ScFv by the use of the IL-22RA1-beta-cell knockouts.

5. Have the β cell-specific IL22RA1 KO mice been described before? Does the deletion have an effect on insulin secretion under steady-state and during HFD? This would emphasize the physiological role of IL-22.

AR27: The reviewer is correct, the IL-22RA1-beta-cell conditional knockouts have not been described previously. Our manuscript, highlighting the endogenous role of IL-22RA1 in the beta cells is currently under 2nd review at Nature Communications (# NCOMMS-23-57747A - Pancreatic Beta-Cell IL-22 Receptor Deficiency Induces Age-Dependent Dysregulation of Insulin Biosynthesis and Systemic Glucose Homeostasis"). We have requested to publish these papers back-to-back for the benefit of the readers; however, we leave it up to the Editors to make this final decision.

6. AST/ALT and systemic metabolism (GTT/ITT) are not assessed in the NASH models (CDAHFD, TAA/HFD and HFD/MCD). These should be presented as they are in vivo measures that are closer to clinical markers that are used in humans.

AR28: We agree that the AST/ALT ratio is an important clinical parameter. We did observe difference in the AST/ALT ratio in the high fat diet model (Fig 3e). However, in the fibrosis models, we have focussed more on the histopathology, which was scored by a clinical pathologist, and demonstrated that IL-22-ScFv clinically improved fibrosis. Fibrosis mouse models do not fully recapitulate the metabolic effects seen in human MASH, and AST/ALT measurements have been reported to overlap between control and fibrotic mice and therefore were not conducted.

7. The main message of the paper is the effect of IL-22 on the liver. Figures 5 and S13 show the effect of IL-22 on HEPG2 in vitro, yet it feels like the experiments are limited and not fully exploited (and it seems a bit disconnected that the adipocyte experiments ended up in the main figure 5f/g). In S13a, what is the effect of IL22-ScFv on HEPG2 lipid content/inflammation? What are the levels of IL-22RA1 expression in each cell type in the liver? What is the effect of IL22-ScFv on stellate cell transcriptomic profile? Are fibrotic genes affected?

AR29: We have now moved the 3T3-adipocyte experiments to supplementary data. In order to demonstrate the effect of IL-22-ScFv on lipids accumulated in hepatocytes, we have conducted additional experiments. These have been added to Fig 5g, h and demonstrate that IL-22-ScFv drives a program to reduce accumulation of lipids in hepatocytes. Changes in overall protein (Bodipy and ACC1) are now added. The molecular changes in stellate cells (LX2) post IL-22-ScFv are shown in Figure 5j, demonstrating a decline in fibrotic gene signature. Please also refer to comment AR05, AR19.

Minor points :

1. Figure 1a and S1a are showing the same schematic, although with different HFD duration.

AR30: We have now corrected the schematic.

2. Figure 2: pictures are too small, readers cannot see labeling.

AR31: We have now removed the DAPI channel only staining to increase the size of the confocal images within the figure.

3. Authors should write mL-22 or hIL-22 in the graphs as well for clarity.

AR32: This has now been added throughout the manuscript for clarity. Thank you.

4. Figure 2: Labelling is wrong for g and h.

AR33: This has now been corrected, thank you.

5. Figure 2h: authors should specify that treated mice are on HFD only.

AR34: We have now labelled the x-axis for clarity.

6. Line 176: Text should refer to Figure 2b and c.

7. Not all graphs/pictures are mentioned in the text (e.g. Fig. 3a and b).

8. Figure S9: Legends are wrong.

AR35: These have now been corrected.

9. Line 243: Pck2 is an enzyme that regulates insulin secretion in beta cells not in liver.

AR36: Please also refer to AR18, we have now highlighted the role of Pck2 in liver.

10. Figure 4b: why using IL22-Fc and not hIL22 as a comparison here?

AR37: We have utilised IL-22Fc as a comparator, as this is the version in clinical trials currently and being advanced for several indications.

11. Figure 5: Not clear if mice are treated ip or subcutaneously? S.c.

AR38: These mice were treated subcutaneously; we have now clarified this in the legend.

12. Figure 1f-l showing adverse effects should be put together with Figure S10.

AR39: These are two different treatment regimens. Fig 1 demonstrates the effects of a daily 8 week treatment with IL-22-Fc and is a toxicity study. Figure S10 is a biweekly treatment with a dose comparable to IL-22-ScFv, which shows efficacy in the HFD model. We have highlighted this in the legends.

REVIEWER COMMENTS

Reviewer #1 (Remarks to the Author):

Accept

Reviewer #2 (Remarks to the Author):

Regarding AR12 in the response to reviewers: The reviewer is trying to inquire whether phenomaster data in supplemental figure 1d took into account differences in body weight. Although VO₂ was not affected, the mice weighed less, which can affect the amount of oxygen consumed. Was ANCOVA used to evaluate the data.

Line 120-129: How does this exclude a direct effect on adipose tissue?

AR19 and lines 324-342. The results in vivo and in vitro clearly show a robust effect on hepatic steatosis, but the mechanistic reason that this occurs is still unclear and that data that are shown confuse the issue even further.

CPT1A is increased in vitro, but tends to be reduced in vivo (Figure 3f). While increased fat oxidation could explain the phenotype, these disparate data and narrow focus on one fat oxidation gene affect the reviewers confidence that this is the mechanism.

ACACA mRNA is increased, but the protein encoded by ACACA (ACC1) is reduced? ACC1 protein was quantified by immunostaining, which is not very quantitative and the reduction shown seems to be less than 5%. It is hard to believe that this is biologically meaningful. Further, the paper cited to justify that suppressing ACC1 would lead to lipid accumulation used genetic deletion of ACC1 and ACC2 due to an unexpected reduction in fatty acid oxidation. Several other papers have use ACC inhibitors to show a reduction in liver fat. All in all the findings with ACC are confusing, contradicting, and don't really fit the observed effects on steatosis.

Similarly, AR20 makes very little metabolic sense and the authors justify their conclusion by citing a study looking at Chrebp knockdown in mice lacking G6pc, which have chronic activation of Chrebp, and is a model of glycogen storage disease. Several other papers have shown that Chrebp deficiency protects mice from hepatic steatosis in models of obesity

(PMID: 16705063). The reviewer is confused by the statement that Chrebp can “reduce concentrations of VLDL-triglycerides and low-density lipoprotein TGs”. Are the authors suggesting that clearance of liver lipid is due to increased secretion of VLDL from the liver? Shouldn't this be reflected by increased blood lipids in the mice?

Overall, the metabolic mechanisms are confusing, confounding, or make little metabolic sense. It would likely require some sort of systematic approach to understand the metabolic mechanism better.

Reviewer #3 (Remarks to the Author):

The authors addressed all my concerns, and the manuscript is clear and strengthened. I recommend acceptance without further revision.

We appreciate the acceptance of our revisions by Reviewers 1 and 3. Thank you to reviewer 2 for their additional queries, we have now addressed these below (Author responses; AR).

1. Regarding AR12 in the response to reviewers: The reviewer is trying to inquire whether phenomaster data in supplemental figure 1d took into account differences in body weight. Although VO₂ was not affected, the mice weighed less, which can affect the amount of oxygen consumed. Was ANCOVA used to evaluate the data.

AR01. Yes, the body weight was taken into account. As body weight is a confounding variable and can influence the VO₂, ANCOVA was used to adjust for the influence of this, no changes in VO₂ are observed with IL-22 treatment. The graph in **Supplementary Fig 1f** also shows the VO₂ when compared to the body mass of the animals. We have now clearly stated this in the figure legend.

2. Line 120-129: How does this exclude a direct effect on adipose tissue?

AR02. These data do not exclude a direct effect on adipose tissue. We have now changed the wording to clarify this in the text.

Old text: We have previously demonstrated rmlL-22 treatment reduced body fat content, which maybe be solely explained by the changes in satiety^{5,8}. Therefore to exclude a direct effect of IL-22 on fat, we assessed the changes in adipocyte size following rmlL-22 treatment, **Fig. 1a**. *IL-22ra1* expression was confirmed in epididymal and subcutaneous fat (**Supplementary Fig 2a**), confirming the possibility of a direct effect.

New text: We have previously demonstrated rmlL-22 treatment reduced body fat content, which maybe be solely explained by the changes in satiety^{5,8}. Therefore, to assess whether IL-22 could directly impact fat, we determined the changes in adipocyte size following rmlL-22 treatment, **Fig. 1a**. *IL-22ra1* expression was confirmed in epididymal and subcutaneous fat (**Supplementary Fig 2a**), confirming the possibility of a direct effect.

3. AR19 and lines 324-342. The results in vivo and in vitro clearly show a robust effect on hepatic steatosis, but the mechanistic reason that this occurs is still unclear and that data that are shown confuse the issue even further. CPT1A is increased in vitro, but tends to be reduced in vivo (Figure 3f). While increased fat oxidation could explain the phenotype, these disparate data and narrow focus on one fat oxidation gene affect the reviewers confidence that this is the mechanism.

ACACA mRNA is increased, but the protein encoded by ACACA (ACC1) is reduced? ACC1 protein was quantified by immunostaining, which is not very quantitative and the reduction shown seems to be less than 5%. It is hard to believe that this is biologically meaningful. Further, the paper cited to justify that suppressing ACC1 would lead to lipid accumulation used genetic deletion of ACC1 and ACC2 due to an unexpected reduction in fatty acid oxidation. Several other papers have use ACC inhibitors to show a reduction in liver fat. All in all the findings with ACC are confusing, contradicting, and don't really fit the observed effects on steatosis. Similarly, AR20 makes very little metabolic sense and the authors justify their conclusion by citing a study looking at Chrebp knockdown in mice lacking G6pc, which have chronic activation of Chrebp, and is a model of glycogen storage disease. Several other

papers have shown that Chrebp deficiency protects mice from hepatic steatosis in models of obesity (PMID: 16705063). The reviewer is confused by the statement that Chrebp can “reduce concentrations of VLDL-triglycerides and low-density lipoprotein TGs”. Are the authors suggesting that clearance of liver lipid is due to increased secretion of VLDL from the liver? Shouldn't this be reflected by increased blood lipids in the mice? Overall, the metabolic mechanisms are confusing, confounding, or make little metabolic sense. It would likely require some sort of systematic approach to understand the metabolic mechanism better.

AR03: Thank you to the reviewer for acknowledging the significant reductions in hepatic steatosis observed with IL-22-ScFv treatment across multiple *in vivo* models and the *in vitro* decrease in lipid accumulation. As the reviewer highlights, these changes are robust and reproducible.

We concur that elucidating the precise mechanisms underlying the *in vivo* effects of IL-22-ScFv presents a challenge due to its likely pleiotropic downstream effects on transcription of multiple genes and pathways. Additionally, the *in vitro* gene expression alterations we identified (Supplementary Figure 12b) do not encompass all the variations contributing to the decrease in lipid content observed in Palmitate-HEPG2 cells (Fig 5f, g).

We agree with the reviewer that the discussion should not solely revolve around changes in individual proteins such as CPT1a, ACC1/2, and Chrebp. Consequently, the detailed discussion on these specific proteins has been omitted from the text. In response to the reviewer's feedback, we have adopted a comprehensive approach to unravel the intricate metabolic alterations underlying the reduction in lipid accumulation observed with IL-22-ScFv. This included conducting mass-spectrometry proteomic analyses on Palmitate HEPG2 cells, both with and without IL-22-ScFv treatment. Notably, this unbiased analysis led to the identification of 5480 proteins with high confidence, of which 250 were differentially regulated by IL-22-ScFv. As anticipated, IL-22 levels were elevated in the IL-22-ScFv-treated samples, providing an internal control (see Figure 1 below; also included as Sup Fig 12c).

While acknowledging the extensive dataset of 250 differentially regulated proteins, it is beyond the scope of our current study to delve into each individually. Hence, we focused our analysis on the top 100 upregulated and top 100 downregulated proteins. Through STRING analyses, we present these findings in Figure 2 below. Our DAVID Bioinformatics investigation, particularly emphasizing KEGG pathway enrichment, unveiled that 36% of the differentially expressed proteins are directly linked to metabolic pathways. Of significance is also function within the endoplasmic reticulum (ER), among the upregulated pathways known to be regulated by IL-22-ScFv. Noteworthy is that these modified pathways encompass diverse facets of lipid and fatty acid metabolism, indicating a comprehensive impact on the cellular metabolic landscape (refer to Fig 3 below; incorporated into Figure 5h).

In summary, this new analysis demonstrates a complex modulation of metabolic pathways that would require much further experimental work to ascertain relative importance of individual proteins.

Figure 1

Figure 3

REVIEWERS' COMMENTS

Reviewer #2 (Remarks to the Author):

My prior concerns have been addressed